# Long-term consequences of the absence of leptin signaling in early life

Angela M Ramos-Lobo[1], Pryscila DS Teixeira[1], Isadora C Furigo[1], Helen M Melo[2], Natalia de M Lyra e Silva[2,3], Fernanda G De Felice[2], Jose Donato Jr[1]*

[1]Department of Physiology and Biophysics, Institute of Biomedical Sciences, University of São Paulo, São Paulo, Brazil; [2]Institute of Medical Biochemistry Leopoldo de Meis, Federal University of Rio de Janeiro, Rio de Janeiro, Brazil; [3]Centre for Neuroscience Studies, Department of Psychiatry, Queen's University, Kingston, Canada

**Abstract** Leptin regulates energy balance and also exhibits neurotrophic effects during critical developmental periods. However, the actual role of leptin during development is not yet fully understood. To uncover the importance of leptin in early life, the present study restored leptin signaling either at the fourth or tenth week of age in mice formerly null for the leptin receptor (LepR) gene. We found that some defects previously considered irreversible due to neonatal deficiency of leptin signaling, including the poor development of arcuate nucleus neural projections, were recovered by LepR reactivation in adulthood. However, LepR deficiency in early life led to irreversible obesity via suppression of energy expenditure. LepR reactivation in adulthood also led to persistent reduction in hypothalamic *Pomc*, *Cartpt* and *Prlh* mRNA expression and to defects in the reproductive system and brain growth. Our findings revealed that early defects in leptin signaling cause permanent metabolic, neuroendocrine and developmental problems.

DOI: https://doi.org/10.7554/eLife.40970.001

*For correspondence:
jdonato@icb.usp.br

Competing interests: The authors declare that no competing interests exist.

## Introduction

The hormone leptin is best known as a key signal to the brain conveying information about body energy reserves (*Maffei et al., 1995*). Consequently, leptin receptor (LepR)-expressing neurons are apt to modulate food intake and energy expenditure according to the amount of energy stored and the current energy balance (*Ramos-Lobo and Donato, 2017*). Hence, null mutations in the genes that encode for leptin or LepR impair the capacity of the hypothalamus to regulate energy homeostasis, leading to profound metabolic dysfunctions in humans and rodents, including severe obesity, hyperphagia, low-energy expenditure, hypoactivity, insulin resistance and infertility (*Halaas et al., 1995*; *Montague et al., 1997*; *Farooqi et al., 2002*; *Licinio et al., 2004*).

In addition to the role of leptin in energy homeostasis, other studies have shown that leptin can affect developmental processes, especially in the brain (*Ahima et al., 1999*). Regarding this aspect, rodents exhibit a postnatal leptin surge between the first and second week of age that is not associated to changes in fat mass or food intake (*Ahima et al., 1998*). In addition, 2-week-old mice do not respond to the anorexigenic effects of leptin (*Mistry et al., 1999*). Thus, leptin does not seem to be involved in the regulation of energy balance during the first weeks of life in mice. However, synaptic plasticity in the hypothalamus is regulated by leptin (*Pinto et al., 2004*). Leptin signaling is also required for the formation of the neural projections from the arcuate nucleus (ARH) to post-synaptic targets, a process that takes place in the first 2 weeks of life (*Bouret et al., 2004a*; *Kamitakahara et al., 2018*). In accordance, leptin promotes neurite outgrowth in vitro and leptin-deficient (*Lep^{ob/ob}*) mice show disruption of ARH neural projections (*Bouret et al., 2004b*).

**eLife digest** Leptin is a hormone that keeps us healthy in many ways. It regulates our body weight by reining in our appetite and fine-tuning the energy we burn, and it helps us establish and maintain our fertility. It also participates in brain development. Leptin performs these roles by attaching to specific receptors in nerve cells and relaying relevant information to the brain.

Early events can trigger life-long changes in the way our body works, a process called metabolic programming. Leptin is believed to participate in this reprogramming mechanism, but its role remains uncertain. In particular, it is still unclear which leptin-driven changes are permanent, and which ones are reversible. Being able to distinguish between the two types of alterations would help to better grasp the role leptin plays in early development.

Here, Ramos-Lobo et al. examined genetically engineered mice born without a working leptin receptor. These animals were impervious to the effects of leptin. Then, once the rodents were adults, they were treated with a drug that restored their leptin receptors, making them sensitive to the hormone again. These experiments revealed that mice without leptin receptors during early life developed obesity, were less able to lose weight and burned less energy. Their reproductive success was also compromised. Finally, the lack of leptin during development caused permanent reduction of the animals' brains, and changes in the activity of certain genes in the organ.

The work by Ramos-Lobo et al. indicates that in mice, lacking leptin sensibility early in life conditions the body to permanently become 'thrifty', burning less energy and making it harder to lose weight. It is rare for humans to be born completely without leptin activity. Yet, having too much or too little food as a baby affects the level of the hormone, or our sensitivity to it: this may permanently change the way our bodies manage energy. Ultimately, learning more about these mechanisms could help us ward off or treat obesity.

DOI: https://doi.org/10.7554/eLife.40970.002

Importantly, exogenous leptin rescues the development of ARH projections of *Lep^{ob/ob}* mice only when provided in the neonatal period. Thus, these findings suggest that the lack of leptin signaling in the first weeks of life permanently disrupts these projections in mice (*Bouret et al., 2004b*).

Important implications arise from the developmental effects of leptin. For example, changes in leptin signaling during development could then affect the organization of critical neural circuits that regulate energy homeostasis, favoring the incidence of metabolic diseases later in life. Thus, leptin may also be involved in the developmental programming (*Ralevski and Horvath, 2015*). Accordingly, early postnatal leptin blockage in rats leads to long-term leptin resistance and susceptibility to diet-induced obesity (*Attig et al., 2008*). Furthermore, neonatal leptin treatment reverses the developmental programming caused by maternal undernutrition (*Vickers et al., 2005*). However, it is worth mentioning that despite the aforementioned evidence about leptin's developmental effects, exogenous leptin replacement can reverse key metabolic abnormalities exhibited by leptin-deficient mice or humans (*Pelleymounter et al., 1995*; *Ioffe et al., 1998*; *Farooqi et al., 2002*; *Licinio et al., 2004*; *Donato et al., 2011*). Therefore, the actual importance of leptin signaling during development for the long-term energy homeostasis and other aspects regulated by leptin is not yet fully understood. Thus, to uncover the role of leptin during development, we investigated the long-term consequences of the absence of leptin signaling in early life. For this purpose, we studied knockout mice for the *Lepr* gene that grew without leptin's effects. However, LepR expression was restored either at the fourth or tenth week of life. Our findings revealed that some defects previously considered irreversible due to the lack of leptin signaling in the neonatal period, such as the poor development of ARH neural projections, can be recovered by LepR reactivation in adulthood. On the other hand, the absence of leptin signaling in early life permanently impairs energy homeostasis, the melanocortin system, as well as the reproduction and brain development.

# Results

## LepR reactivation in adult mice

To study the importance of leptin signaling during development, we used the LepRNull mouse model that carries a transcription blocker between exons 16 and 17 of the *Lepr* gene (*Berglund et al., 2012*), preventing the expression of the long LepR isoform (*Lepr-b*). As expected, adult male and female LepRNull mice were morbidly obese and hyperphagic (*Figure 1A–D*). Since the transcription blocker of LepRNull mice is flanked by *LoxP* sites, a temporal Cre expression can permanently remove this cassette. LepRNull mice were bred with animals expressing Cre-ERT2 fusion protein under the human ubiquitin C promoter sequence. Thus, tamoxifen injections were able to induce Cre recombinase activity ubiquitously, restoring LepR expression in Ubi-LepRNull mice. We decided to induce LepR reactivation when mice were 10 weeks old because the vast majority of developmental processes were completed at this age. We observed that tamoxifen injections caused no long-term changes in body weight or food intake in lean controls (Ubi group) and LepRNull mice (*Figure 1A–D*). In contrast, Ubi-LepRNull mice exhibited a sustained weight loss and marked food intake reduction in the first 4 weeks after tamoxifen treatment (*Figure 1A–D*).

To confirm the gene reactivation, we assessed the hypothalamic expression of *Lepr-b* mRNA and observed the expected suppression in LepRNull mice (0.01 ± 0.00 a.u.; *n* = 8), compared to lean controls (1.00 ± 0.06 a.u.; *n* = 8). Importantly, LepR reactivation recovered the expression of *Lepr-b* mRNA in Ubi-LepRNull mice (0.66 ± 0.04 a.u.; *n* = 8; p=0.071 vs. Ubi mice). To further validate our experimental model, we evaluated the leptin responsiveness in key hypothalamic nuclei that contain LepR-expressing neurons (*Ramos-Lobo and Donato, 2017*). For this purpose, we assessed the capacity of leptin to induce the phosphorylation of STAT3 (pSTAT3), which depends on the long LepR isoform (*Bates et al., 2003*). As expected, an i.p. leptin injection induced a robust pSTAT3 immunoreactivity in the ventromedial subdivision of the ARH (ARHvm), lateral subdivision of the ARH (ARHl), dorsomedial nucleus (DMH), ventromedial nucleus (VMH) and ventral premammillary nucleus (PMv) of control Ubi mice (*Figure 1E,H*). In accordance with the deficiency of the long LepR isoform, LepRNull mice show virtually no pSTAT3 in the hypothalamus (*Figure 1F,I*). Importantly, tamoxifen injections restored leptin responsiveness in Ubi-LepRNull mice to a similar extent as that found in the Ubi group (*Figure 1G,J,K*). To functionally assess leptin sensitivity, we determined the acute anorexigenic effects caused by an i.p. leptin injection. Leptin induced a similar decrease of food intake and body weight in Ubi and Ubi-LepRNull mice, whereas LepRNull mice remained unresponsive to leptin (*Figure 1L*). Altogether, these findings demonstrate that although mice were depleted from the effects of leptin until the 10[th] week of life, tamoxifen injections completely restored leptin signaling in adult Ubi-LepRNull mice.

## Absence of leptin signaling in early life alters the energy balance and predisposes the animals to obesity

We found that body weight and food intake of Ubi-LepRNull mice were stabilized 6 weeks after tamoxifen treatment (*Figure 1A–D*). Therefore, all remaining analyses were performed after this period. Despite the marked weight loss induced by LepR reactivation, Ubi-LepRNull male mice remained heavier and had higher body adiposity and leptinemia, compared to control Ubi mice (*Figure 2A–C*). These changes occurred despite a complete normalization of food intake in Ubi-LepRNull mice (*Figure 2D*). On the other hand, oxygen consumption ($VO_2$) remained significantly lower in Ubi-LepRNull male mice compared to lean controls (*Figure 2E,F*), indicating that the increased adiposity of Ubi-LepRNull mice could be explained by a partial recovery in their energy expenditure. The respiratory exchange ratio (RER) of Ubi-LepRNull male mice was normalized, restoring the circadian rhythm of fuel oxidation which was absent in LepR deficient mice (*Figure 2G,H*). As seen in the energy expenditure, the phenotype of hypoactivity observed in LepRNull mice was only partially restored in Ubi-LepRNull male animals (*Figure 2I,J*).

The energy homeostasis was also evaluated in females. As seen in males, Ubi-LepRNull females remained heavier and had higher body adiposity compared to Ubi mice (*Figure 2—figure supplement 1A,B*), although serum leptin levels and food intake were completely normalized (*Figure 2—figure supplement 1C,D*). Ubi-LepRNull females also exhibited a partial recovery in their energy expenditure in the dark phase compared to Ubi mice (*Figure 2—figure supplement 1E,F*).

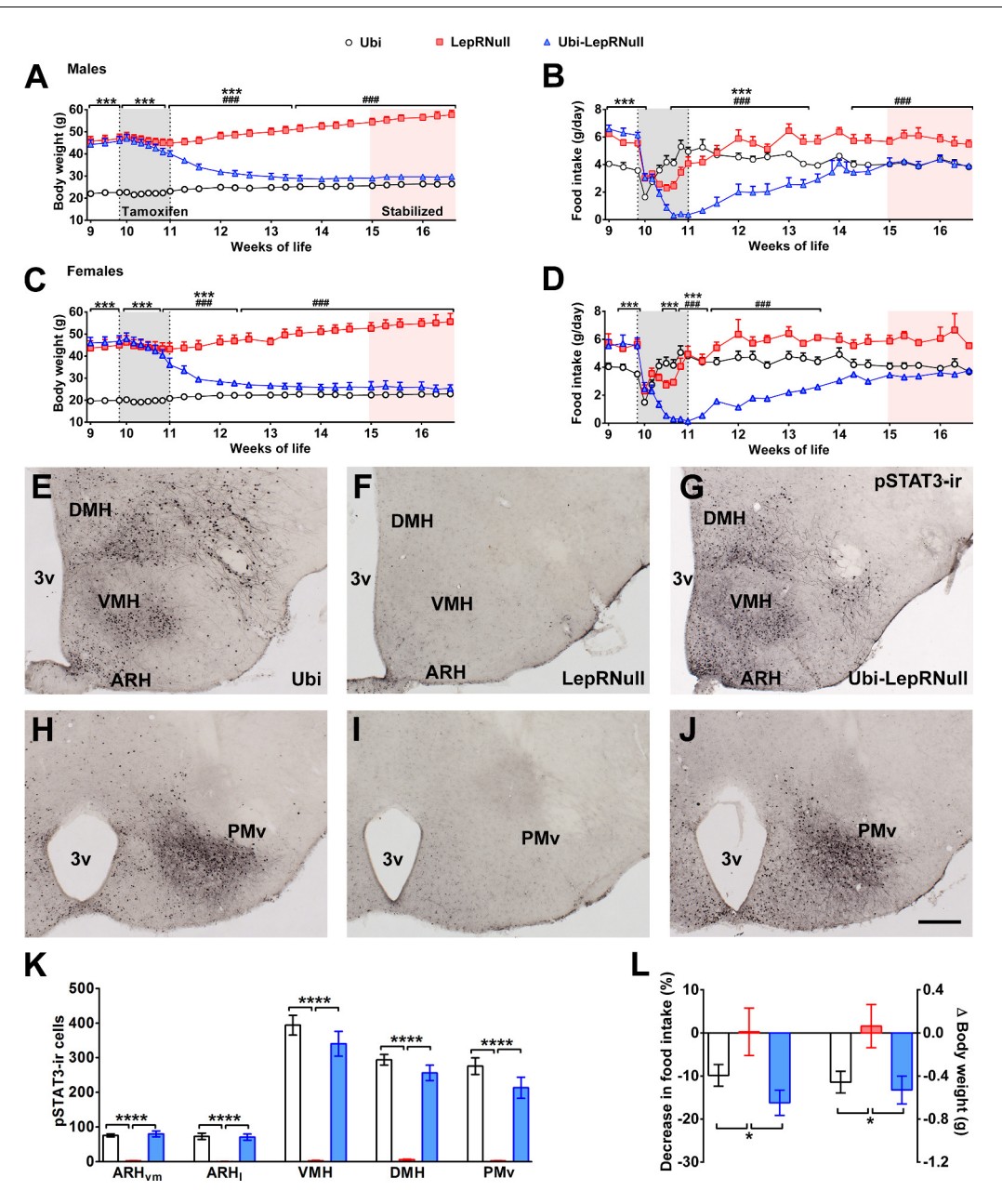

**Figure 1.** Validation of LepR reactivation in 10-week-old mice. (**A**) Body weight of male mice (main effect of Genotype [$F_{(2, 23)}$=153.7, p<0.0001], main effect of Time [$F_{(24, 552)}$=22.37, p<0.0001] and interaction [$F_{(48, 552)}$=105.5, p<0.0001]; n = 6–12) before, during and after tamoxifen treatment at 10 weeks of age. (**B**) Food intake of male mice (main effect of Genotype [$F_{(2, 23)}$=14.13, p<0.0001], main effect of Time [$F_{(23, 529)}$=29.30, p<0.0001] and interaction [$F_{(46, 529)}$=18.58, p<0.0001]; n = 5–12). (**C**) Body weight of female mice (main effect of Genotype [$F_{(2, 19)}$=154.1, p<0.0001], main effect of Time [$F_{(24, 456)}$=23.75, p<0.0001] and interaction [$F_{(48, 456)}$=114.5, p<0.0001]; n = 2–16). (**D**) Food intake of female mice (main effect of Genotype [$F_{(2, 15)}$=3.779, p=0.0469], main effect of Time [$F_{(23, 345)}$=8.386, p<0.0001] and interaction [$F_{(46, 345)}$=5.435, p<0.0001]; n = 1–13). ### p<0.0001 Ubi-LepRNull vs LepRNull mice. ***p<0.0001 Ubi-LepRNull vs Ubi mice. (**E–J**) Brightfield photomicrographs of pSTAT3 immunoreactive neurons in Ubi (**E, H**), LepRNull (**F, I**) and Ubi-LepRNull mice (**G, J**) after an acute leptin injection (5 µg/g b.w.). 3 v, third ventricle. Scale Bar = 200 µm. (**K**) Number of pSTAT3 cells after an i.p. leptin injection (n = 5–6). ***p<0.001. (**L**) Acute changes in food intake (left panel) ($F_{(2, 28)}$=5.927, *p=0.0071, n = 8–13) and body weight (right panel) ($F_{(2, 33)}$=4.524, *p=0.0184, n = 8–18) after an i.p. leptin injection (2.5 µg/g b.w.) in comparison to PBS injection (*Figure 1—source data 1*).

DOI: https://doi.org/10.7554/eLife.40970.003

*Figure 1 continued on next page*

*Figure 1 continued*

The following source data is available for figure 1:

**Source data 1.** Data regarding leptin responsiveness.

DOI: https://doi.org/10.7554/eLife.40970.004

Additionally, RER was normalized after LepR reactivation (*Figure 2—figure supplement 1G,H*), whereas Ubi-LepRNull females displayed a partial recovery in their ambulatory activity, compared to Ubi mice (*Figure 2—figure supplement 1I,J*).

## Metabolic challenges confirm the thrifty phenotype induced by the lack of leptin signaling early in life

Food deprivation produces compensatory decreases in energy expenditure as a defense mechanism against weight loss (*Leibel et al., 1995*). Thus, we subjected the animals to 24 hr fasting to evaluate their metabolic responses during a situation of negative energy balance (*Figure 3A–C*). LepRNull and Ubi-LepRNull mice showed a lower fasting-induced weight loss compared to Ubi mice (*Figure 3A*). Notably, while Ubi mice adapted to fasting by decreasing the energy expenditure, both LepRNull and Ubi-LepRNull mice exhibited an attenuated reduction compared to baseline (*Figure 3B,C*). Overfeeding can also produce compensatory responses in humans and mice in order to prevent excessive weight gain (*Leibel et al., 1995*). We measured the food intake 4, 12, 24 and 48 hr after a 24 hr fasting period and observed a similar refeeding response between male and female Ubi and Ubi-LepRNull mice, whereas male LepRNull mice exhibited an increase in food intake after 24 and 48 hr of refeeding (*Figure 3—figure supplement 1A,B*). Additionally, we evaluated the metabolic responses produced by a high-fat diet (HFD) intake for 48 hr. Interestingly, both LepRNull and Ubi-LepRNull mice showed higher HFD intake compared to Ubi animals (*Figure 3D*). However, HFD-induced thermogenesis was similar among the experimental groups (*Figure 3E,F*).

## Absence of leptin signaling in early life affects the central melanocortin system and the hypothalamic expression of enzymes related to epigenetic changes

To investigate the possible causes of the metabolic imbalances exhibited by Ubi-LepRNull mice, we assessed the hypothalamic expression of key genes involved in energy balance regulation. As expected, LepRNull mice showed increased agouti-related protein (*Agrp*) and neuropeptide Y (*Npy*) mRNA levels in the hypothalamus, whereas the expression of proopiomelanocortin (*Pomc*) and cocaine and amphetamine regulated transcript (*Cartpt*) was reduced compared to Ubi mice (*Figure 3G*). LepR reactivation downregulated *Agrp* and *Npy* mRNA levels in the hypothalamus of Ubi-LepRNull mice. However, *Pomc* and *Cartpt* expression was not restored and remained significantly lower compared to control Ubi mice (*Figure 3G*). To assess whether the absence of leptin signaling in early life could have affected the viability of POMC neurons, we determined the number of α-melanocyte-stimulating hormone (α-MSH) or β-endorphin immunoreactive cells in the ARH (*Figure 3—figure supplement 2*). We observed a similar number of ARH POMC neurons between Ubi and Ubi-LepRNull mice (*Figure 3—figure supplement 2*).

Neurons outside the ARH also regulate the thermogenic effect of leptin, including those that express the RFamide prolactin-releasing peptide (*Prlh*) in the DMH (*Dodd et al., 2014*). Thus, we assessed the hypothalamic expression of *Prlh* mRNA, which was suppressed in LepRNull mice and remained lower in the hypothalamus of Ubi-LepRNull mice (*Figure 3G*). Epigenetic changes in early life can cause long-term consequences in energy homeostasis and increase the risk of obesity in adulthood (*Lillycrop and Burdge, 2011*). Hence, the expression of some histone deacetylases (*Hdac*) was also determined and we observed that *Hdac3* and *Hdac5* mRNA levels were suppressed in the hypothalamus of LepRNull mice and were not normalized after LepR reactivation (*Figure 3G*). *Hdac8* expression was increased in the hypothalamus of LepRNull mice, but it returned to normal values in Ubi-LepRNull mice (*Figure 3G*). DNA methyltransferase (*Dnmt*) enzymes catalyze the transfer of a methyl group to DNA, affecting many biological functions including the energy homeostasis via hypothalamic neurons (*Kohno et al., 2014*). We analyzed the expression of some *Dnmt* and we observed that *Dnmt3a* and *Dnmt3b* mRNA levels were suppressed in the hypothalamus of both

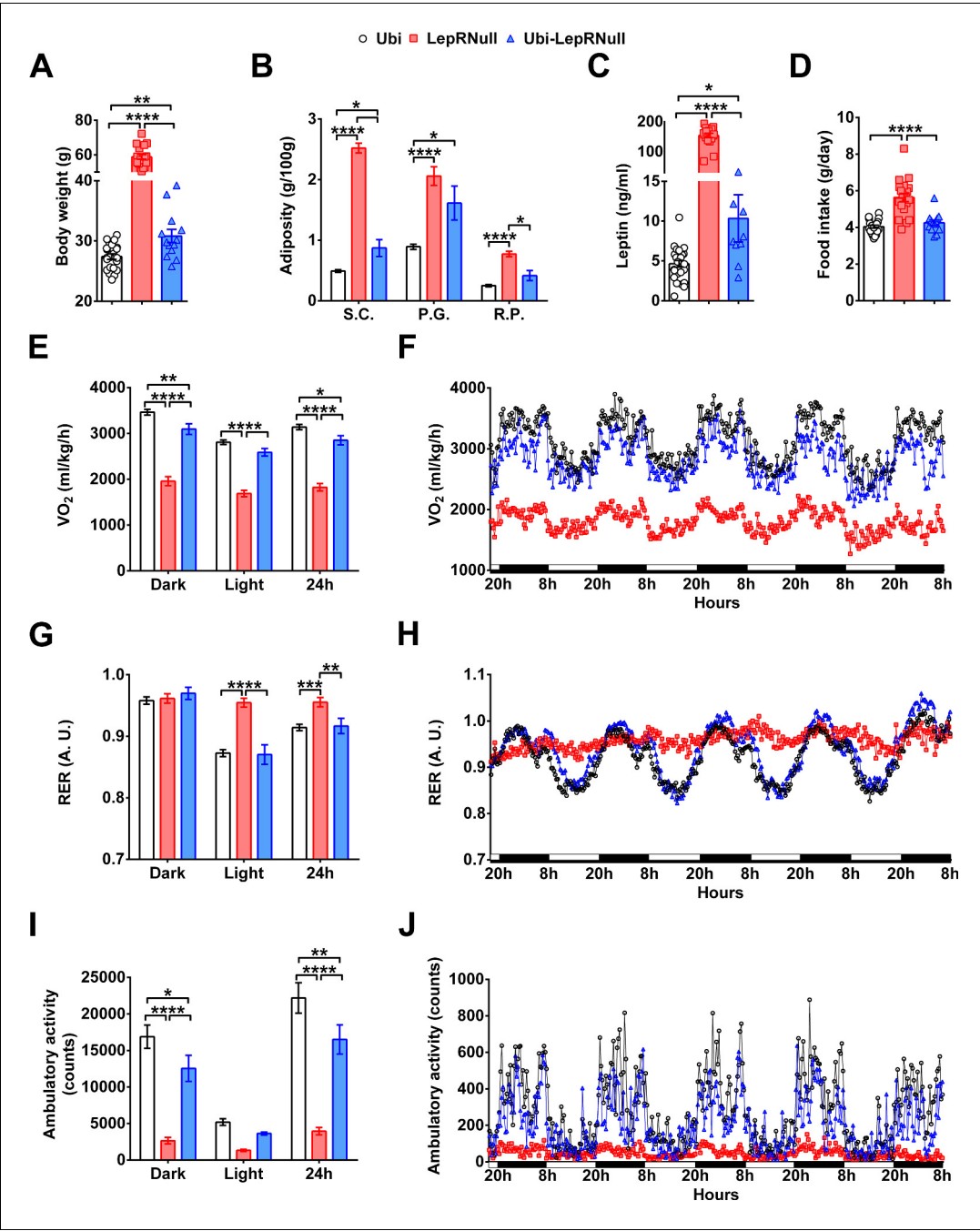

**Figure 2.** Absence of leptin signaling in early life alters the energy balance and predisposes the animals to obesity. (A–D) Body weight (*n* = 13–22), body adiposity (S.C., subcutaneous; P.G., perigonadal; R.P., retroperitoneal fat pads; *n* = 10–25), serum leptin concentration (*n* = 9–25) and food intake (*n* = 13–22) in male mice treated with tamoxifen at 10 weeks of age. (E–F) Energy expenditure (VO$_2$) during dark phase, light phase and 24 hr in male mice (*n* = 9–15). (G–H) Respiratory exchange ratio (RER) in male mice (*n* = 9–15). (I–J) Voluntary ambulatory activity in male mice (*n* = 9–15). *p<0.05; **p<0.01; ***p<0.001; ****p<0.0001 (*Figure 2—source data 1*).

DOI: https://doi.org/10.7554/eLife.40970.005

The following source data and figure supplements are available for figure 2:

**Source data 1.** Data regarding changes in energy balance in male mice.
DOI: https://doi.org/10.7554/eLife.40970.008

**Figure supplement 1.** LepR reactivation in adult females does not restore completely the energy homeostasis.
*Figure 2 continued on next page*

*Figure 2 continued*

DOI: https://doi.org/10.7554/eLife.40970.006

**Figure supplement 1—source data 1.** Data regarding energy balance in female mice.

DOI: https://doi.org/10.7554/eLife.40970.007

LepRNull and Ubi-LepRNull mice, compared to Ubi animals (*Figure 3G*). No changes among the groups were observed in the hypothalamic expression of *Dnmt1* (*Figure 3G*).

## Absence of leptin signaling in early life causes insulin resistance

Glucose homeostasis was also evaluated in our experimental animals. LepRNull mice showed the expected glucose intolerance, as well as increased insulin resistance and serum insulin concentrations (*Figure 4*). Glucose tolerance of Ubi-LepRNull mice was normalized in both males (*Figure 4A, B*) and females (*Figure 4F,G*). However, Ubi-LepRNull mice remained less responsive to insulin (*Figure 4C,D,H,I*). In accordance with their insulin resistance, serum insulin concentrations remained significantly higher in Ubi-LepRNull mice, compared to Ubi mice (*Figure 4E,J*). Thus, absence of leptin signaling in early life produced long-term alterations in glucose homeostasis.

## LepR reactivation before the onset of obesity confirms the energy imbalance of Ubi-LepRNull mice

LepR reactivation in 10-week-old mice induced a marked weight loss (*Figure 1A,C*). Thus, it is unclear whether the alterations in the energy and glucose homeostasis of Ubi-LepRNull mice were caused by the absence of leptin in early life or were associated with the metabolic consequences of a robust weight loss (*Leibel et al., 1995*). To answer this question, we produced a new group of animals in which tamoxifen treatment started at the fourth week of life (*Figure 5—figure supplement 1*). Therefore, LepR reactivation was induced before the onset of obesity, and consequently Ubi-LepRNull mice were not exposed to weight loss (*Figure 5—figure supplement 1*). Hypothalamic gene expression analysis indicated that Ubi-LepRNull mice exhibited a similar *Lepr-b* mRNA expression (1.26 ± 0.11 a.u.; *n* = 7) compared to Ubi mice (1.00 ± 0.09 a.u.; *n* = 8; p=0.5263), whereas LepRNull mice showed suppressed hypothalamic *Lepr-b* mRNA levels (0.01 ± 0.00 a.u.; *n* = 7). We started the in vivo analyses at the same age of previously studied adult animals (approximately 15 weeks of age). LepR reactivation before the onset of obesity did not affect the body weight of Ubi-LepRNull males in the long-term, but Ubi-LepRNull females still showed higher body weight than Ubi mice (*Figure 5A*). Notably, both male and female Ubi-LepRNull mice exhibited increased fat deposition in the perigonadal depot (*Figure 5B*), although no differences were observed in other fat pads (data not shown) or in serum leptin concentration in male mice (*Figure 5C*). Food intake was also completely normalized in male and female Ubi-LepRNull mice (*Figure 5D*). As seen in adult animals, LepR reactivation before the onset of obesity led to suppressed energy expenditure in the dark phase and over a 24 hr period (*Figure 5E,F*). In contrast, RER and ambulatory activity were normalized in young Ubi-LepRNull male mice (*Figure 5G–J*). Glucose homeostasis was also evaluated in mice that received tamoxifen treatment before the onset of obesity. In contrast to the results observed after LepR reactivation in adult mice, LepR reactivation before the onset of obesity completely normalized the glucose tolerance and insulin sensitivity in both male and female Ubi-LepRNull mice (*Figure 5—figure supplement 2*).

## Lack of leptin signaling in early life disrupts the reproductive system

Leptin not only controls the energy and glucose homeostasis, but it also plays an important role regulating several endocrine systems (*Ahima et al., 1996*; *Ramos-Lobo and Donato, 2017*). Thus, we also evaluated the endocrine consequences of the absence of leptin signaling in early life. We initially assessed the thyroid axis and observed that LepRNull males exhibited the expected suppression in serum T4 concentration (*Ahima et al., 1996*), while Ubi-LepRNull mice showed similar values to Ubi animals (*Figure 6A*). LepRNull mice also displayed higher serum corticosterone concentration compared to Ubi animals, whereas LepR reactivation normalized the hypothalamic-pituitary-adrenal axis (*Figure 6B*).

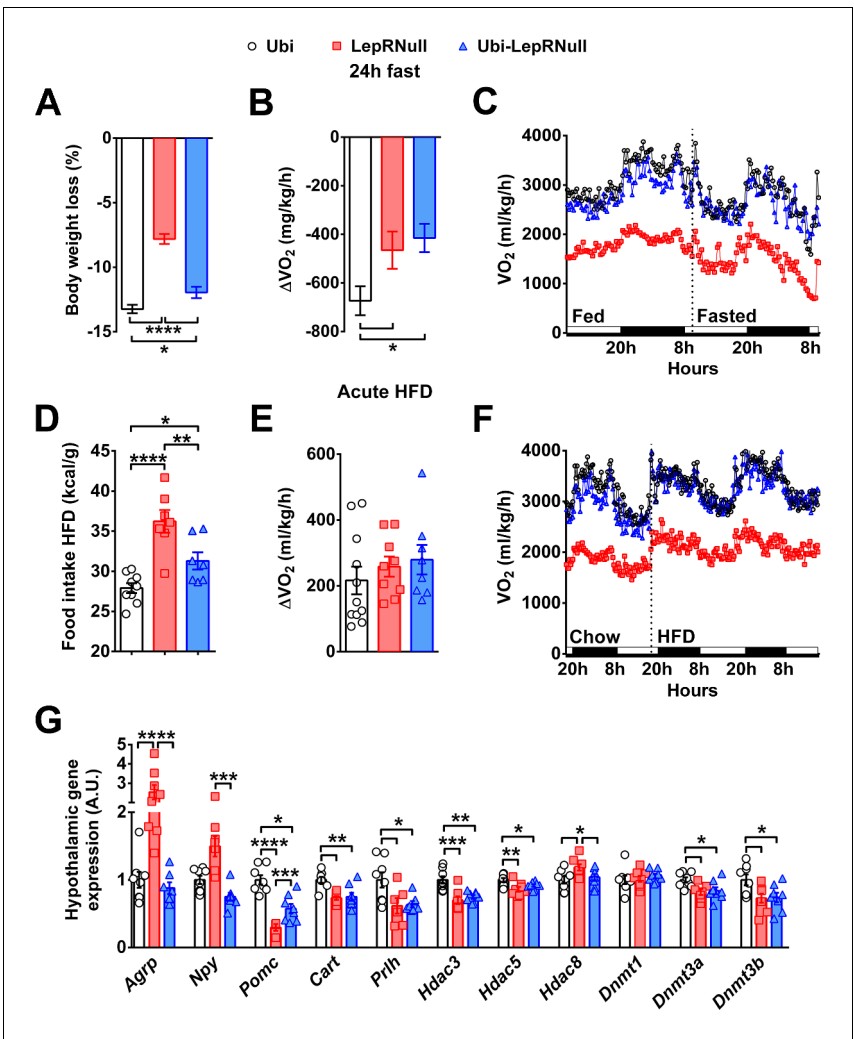

**Figure 3.** Absence of leptin signaling in early life affects the response to negative and positive energy balance, the central melanocortin system and the expression of enzymes related to epigenetic changes. (**A–C**) Variations in % body weight and energy expenditure ($VO_2$) caused by 24 hr fasting compared to *ad libitum* feeding ($n$ = 9–21). (**D–F**) Calorie intake and changes in $VO_2$ caused by the intake of HFD for 48 hr, compared to a regular low-fat diet ($n$ = 7–12). HFD was provided 2 hr before dark phase. (**G**) Hypothalamic mRNA expression in Ubi, LepRNull and Ubi-LepRNull mice ($n$ = 6–8). *p<0.05; **p<0.01; ***p<0.001; ****p<0.0001 (*Figure 3—source data 1* and *Figure 3—source data 2*).

DOI: https://doi.org/10.7554/eLife.40970.009

The following source data and figure supplements are available for figure 3:

**Source data 1.** Data regarding situations of negative and positive energy balance.
DOI: https://doi.org/10.7554/eLife.40970.014
**Source data 2.** Primer list.
DOI: https://doi.org/10.7554/eLife.40970.015
**Figure supplement 1.** Refeeding after fasting.
DOI: https://doi.org/10.7554/eLife.40970.010
**Figure supplement 1—source data 1.** Data regarding food intake during refeeding.
DOI: https://doi.org/10.7554/eLife.40970.011
**Figure supplement 2.** Alterations in metabolism following LepR reactivation in adult mice are not caused by decreased number of POMC cells.
DOI: https://doi.org/10.7554/eLife.40970.012
**Figure supplement 2—source data 1.** Data regarding the number of POMC neurons in the hypothalamus.
DOI: https://doi.org/10.7554/eLife.40970.013

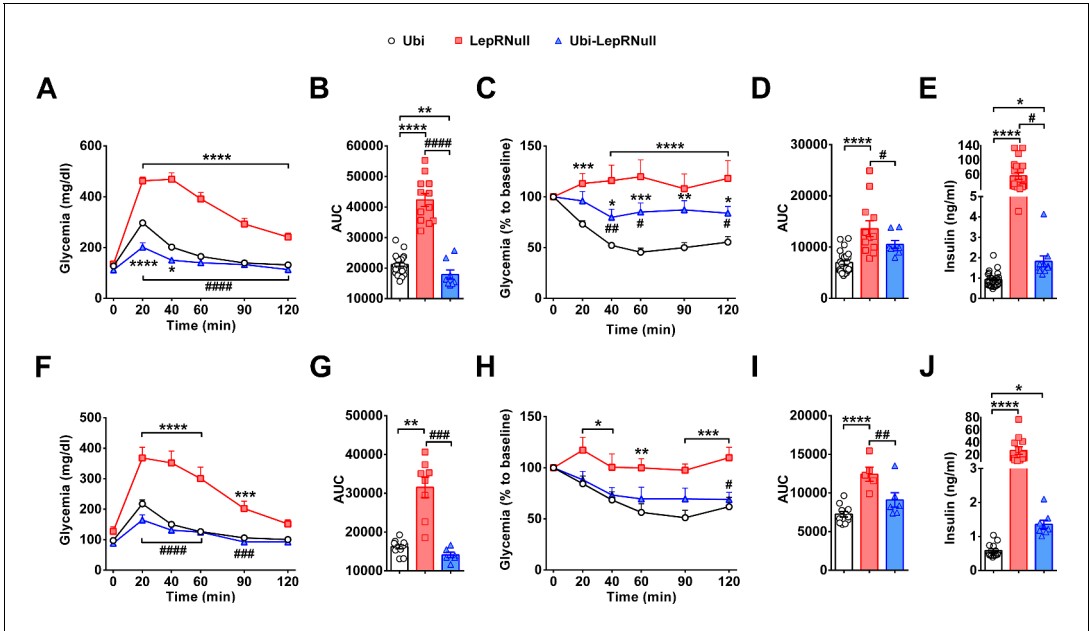

**Figure 4.** Absence of leptin signaling in early life causes insulin resistance. (**A–B**) Glucose tolerance test (GTT; 2 g glucose/kg b.w.; i.p.) and the area under the curve (AUC) of the GTT in male mice ($n$ = 10–32). (**C–D**) Insulin tolerance test (ITT; 1 IU/kg b.w.; i.p.) and the AUC of the ITT in male mice ($n$ = 7–27). (**E**) Serum insulin concentration in male mice ($n$ = 10–25). (**F–G**) GTT and the AUC of the GTT in female mice ($n$ = 6–13). (**H–I**) ITT and the AUC of the ITT in female mice ($n$ = 6–12). (**J**) Serum insulin concentration in female mice ($n$ = 8–14). *$p<0.05$ vs Ubi group; **$p<0.01$ vs Ubi group; ***$p<0.001$ vs Ubi group; ****$p<0.0001$ vs Ubi group. # $p<0.05$ vs LepRNull group; ## $p<0.01$ vs LepRNull group; ### $p<0.001$ vs LepRNull group; #### $p<0.0001$ vs LepRNull group (*Figure 4—source data 1*).

DOI: https://doi.org/10.7554/eLife.40970.016

The following source data is available for figure 4:

**Source data 1.** Data regarding glucose homeostasis.

DOI: https://doi.org/10.7554/eLife.40970.017

Next, the reproductive system was evaluated. In males, LepRNull mice showed lower serum testosterone concentration compared to Ubi animals, whereas Ubi-LepRNull mice exhibited intermediate values (*Figure 6C*). Notably, LepR reactivation in adult animals was unable to completely restore testicle weight, which was reduced by LepR deficiency (*Figure 6D*). To evaluate how these changes affected their fertility, Ubi and Ubi-LepRNull males were mated with wild-type females for 28 days. We observed that only 10% (1 out 10) of females mating with Ubi-LepRNull males got pregnant, whereas the percentage of pregnancy increased to approximately 50% when females mated with Ubi mice.

Leptin deficiency impairs hypothalamic gonadotropin-releasing hormone (GnRH) release, leading to hypogonadotropic hypogonadism (*Chehab et al., 1996*; *Licinio et al., 2004*). Consequently, GnRH peptide content accumulates in the median eminence/mediobasal hypothalamus of food-deprived or LepR-deficient animals (*Polkowska et al., 2006*; *Donato et al., 2011*). We evaluated the density of GnRH fibers reaching the median eminence and observed the expected increase in the integrated optical density of GnRH staining in LepRNull mice, compared to Ubi animals (*Figure 6E,F*). Remarkably, LepR reactivation did not restore GnRH fiber density in the hypothalamus of Ubi-LepRNull mice (*Figure 6G,H*). GnRH release is stimulated by several neurotransmitters, including the kisspeptins (*Messager et al., 2005*) and nitric oxide (*Bellefontaine et al., 2014*). Thus, we evaluated the hypothalamic mRNA expression of *Kiss1* and *Nos1*. We observed a reduction in the expression of these transcripts in the hypothalamus of LepRNull and Ubi-LepRNull mice (*Figure 6I*), indicating that LepR reactivation in adults was not able to normalize *Kiss1* and *Nos1* mRNA levels. We only evaluated the testicle weight in mice that had the LepR reactivation before the onset of obesity. However, as seen in adult animals, LepR reactivation in the fourth week of life was not capable of restoring testicle weight to values found in Ubi mice (*Figure 6J*).

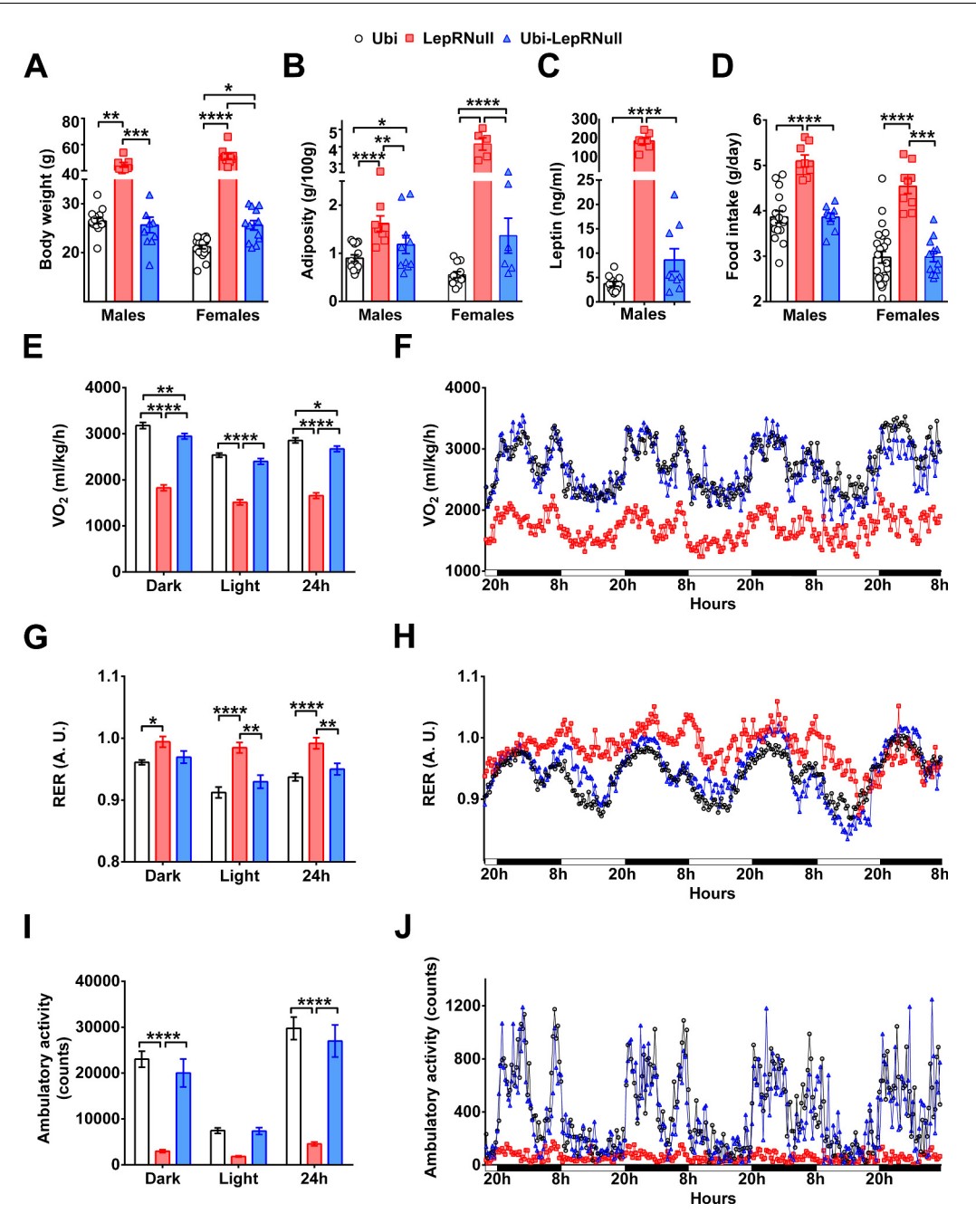

**Figure 5.** LepR reactivation before the onset of obesity confirmed the energy imbalance of Ubi-LepRNull mice. (A–D) Body weight (males, $n$ = 7–16; females, $n$ = 10–24), perigonal fat pad (males, $n$ = 8–16; females, $n$ = 5–12), serum leptin concentration (males, $n$ = 8–12) and food intake (males, $n$ = 8–16; females, $n$ = 5–12) in mice treated with tamoxifen at 4 weeks of age. (E–F) Energy expenditure (VO$_2$) during dark phase, light phase and 24 hr in male mice ($n$ = 8–16). (G–H) Respiratory exchange ratio (RER) in male mice ($n$ = 8–16). (I–J) Voluntary ambulatory activity in male mice ($n$ = 8–16). *p<0.05; **p<0.01; ***p<0.001; ****p<0.0001 (*Figure 5—source data 1*).

DOI: https://doi.org/10.7554/eLife.40970.018

The following source data and figure supplements are available for figure 5:

**Source data 1.** Data regarding energy balance in young mice.
DOI: https://doi.org/10.7554/eLife.40970.023

**Figure supplement 1.** LepR reactivation before the onset of obesity in male and female mice.
DOI: https://doi.org/10.7554/eLife.40970.019

*Figure 5 continued on next page*

*Figure 5 continued*

**Figure supplement 1—source data 1.** Data regarding leptin receptor reactivation in young mice.
DOI: https://doi.org/10.7554/eLife.40970.020
**Figure supplement 2.** LepR reactivation before the onset of obesity completely normalized the glucose tolerance and insulin sensitivity.
DOI: https://doi.org/10.7554/eLife.40970.021
**Figure supplement 2—source data 1.** Data regarding glucose homeostasis in young mice.
DOI: https://doi.org/10.7554/eLife.40970.022

In females, the reproductive system was evaluated through the daily inspection of the vaginal smear. Ubi mice showed the expected variations in the estrous cycle (*Figure 6K*). In contrast, the vaginal cytology of LepRNull mice indicated a prevalence of leukocytes and a complete absence of cornification and estrous cyclicity. LepR reactivation increased the appearance of cornified cells in the vaginal smear, although the disruption of the estrous cycle was still evident in Ubi-LepRNull mice (*Figure 6K*). Thus, these data suggest disruption in the reproductive system of male and female mice that grew without leptin signaling until adulthood.

## Brain development is affected by the absence of leptin signaling in early life, although ARH projections are normalized in Ubi-LepRNull mice

Leptin or LepR deficiency leads to reduced brain mass (*Ahima et al., 1999*), disruption of the projections from ARH neurons to the paraventricular nucleus of the hypothalamus (PVH) (*Bouret et al., 2004a*; *Bouret et al., 2004b*; *Kamitakahara et al., 2018*), as well as behavioral and cognitive problems (*Liu et al., 2011*). Therefore, we evaluated the possible consequences of the lack of leptin signaling in early life on brain development. LepRNull male and female mice exhibited reduced brain mass (*Figure 7A,B*). Notably, LepR reactivation in adult animals partially recovered brain mass, although it remained significantly lower in Ubi-LepRNull mice compared to Ubi animals (*Figure 7A, B*). We then assessed the hypothalamic expression of several neurotrophic factors. LepRNull mice exhibited reduced mRNA levels of *Bdnf*, *Igf1* and *Dlg4*, which encode important neurotrophic factors and proteins involved in synaptic plasticity and memory formation (*Figure 7C*). The expression of none of these transcripts was completely restored in Ubi-LepRNull mice (*Figure 7C*). Additionally, the glial fibrillary acidic protein (*Gfap*) expression was suppressed in the hypothalamus of LepRNull and Ubi-LepRNull mice in comparison with Ubi animals (*Figure 7C*). The brain mass was also determined in mice that recovered LepR expression before the onset of obesity. While Ubi-LepRNull males still exhibited reduced brain mass compared to Ubi mice (*Figure 7D*), no significant difference was observed between Ubi and Ubi-LepRNull females (*Figure 7E*).

To determine whether the aforementioned alterations in Ubi-LepRNull mice could lead to behavioral or cognitive deficits, male mice were subjected to behavioral tests. In the open field, we observed that LepRNull mice showed reduced exploration, whereas LepR reactivation normalized this behavior (*Figure 7—figure supplement 1A*). Next, spatial learning and memory were analyzed in the Barnes maze. In this case, LepRNull mice were not evaluated due to their decreased voluntary ambulatory activity. Along the training period, Ubi and Ubi-LepRNull mice reduced the primary latency (*Figure 7—figure supplement 1B*; p<0.0001) and the number of primary errors (*Figure 7—figure supplement 1C*; p<0.0001) to reach the escape box, indicating that both groups learned the task. During the test sessions, Ubi and Ubi-LepRNull mice also showed similar learning capacity since the time spent in the target zone were similar between the groups 24 hr (*Figure 7—figure supplement 1D,E*) and 1 week (*Figure 7—figure supplement 1F,G*) after the training period.

Previous studies have shown that there is a critical neonatal period, in which leptin plays a neurotrophic role inducing the development of neural projections from ARH neurons to post-synaptic targets, including the PVH (*Bouret et al., 2004b*; *Kamitakahara et al., 2018*). To initially investigate the projections of ARH neurons to the PVH, we crossed $Lep^{ob/+}$ mice with LepR-Cre::LSL-tdTomato mice, in order to produce leptin deficient mice expressing the tdTomato fluorescent protein only in LepR cells. Thus, we compared the distribution of axons in the PVH between control (LepR-Cre::LSL-tdTomato) and $Lep^{ob/ob}$::LepR-Cre::LSL-tdTomato mice. We found a significant reduction in the density of axons from LepR-expressing neurons in the PVH of $Lep^{ob/ob}$ mice compared to control

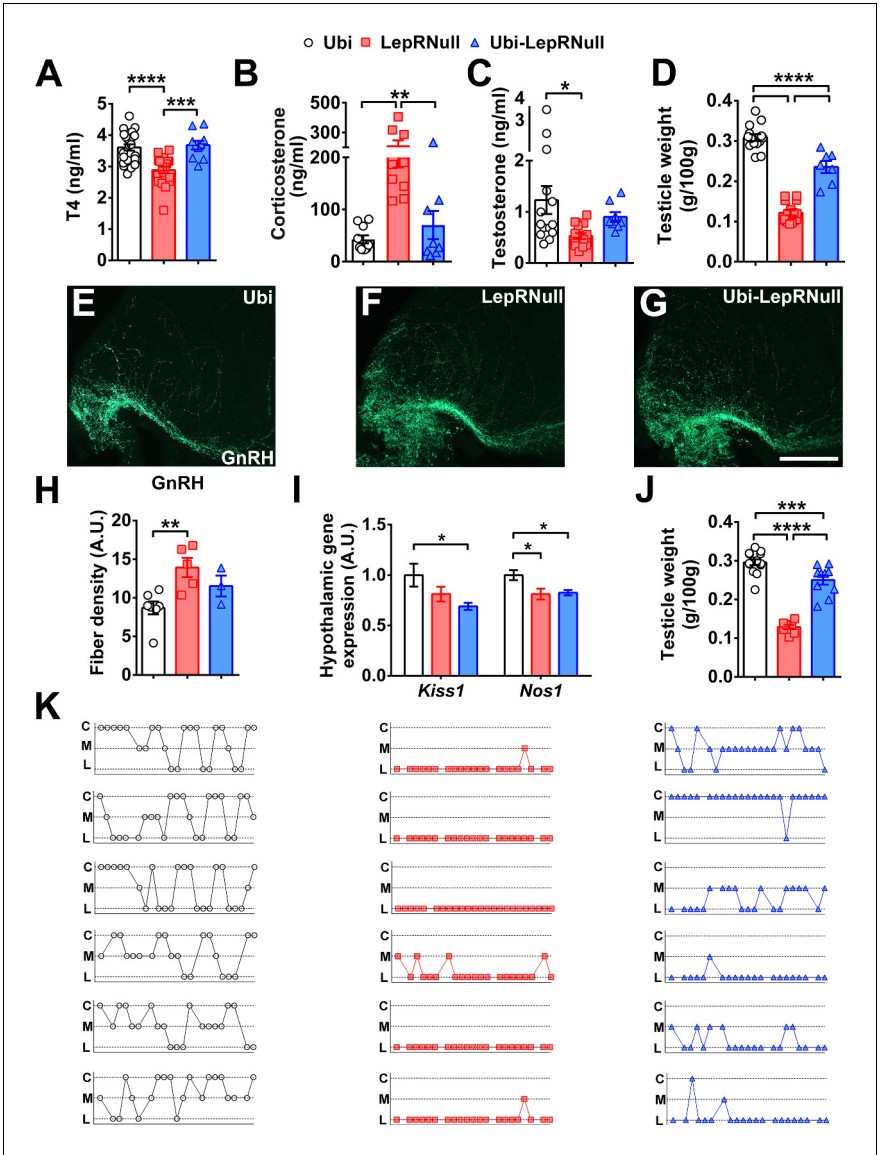

**Figure 6.** LepR reactivation does not completely restore the reproductive axis of adult mice. (A–C) Serum concentration of T4 (*n* = 10–25), corticosterone (*n* = 8–10) and testosterone (*n* = 8–14) in mice treated with tamoxifen at 10 weeks of age. (D) Testicle weight between the groups of adult mice (*n* = 7–15). (E–G) Epifluorescence photomicrographs of immunoreactive GnRH fibers in the mediobasal hypothalamus of Ubi (E), LepRNull (F) and Ubi-LepRNull (G) adult males. Scale Bar = 100 μm. (H) Integrated optical density of GnRH immunoreactive fibers of adult males (*n* = 3–7). (I) Hypothalamic mRNA expression of *Kiss1* and *Nos1* in adult males (*n* = 8). (J) Testicle weight in mice treated with tamoxifen at 4 weeks of age (*n* = 8–14). (K) Representation of the estrous cycle during 25 days determined by the daily inspection of the vaginal smear in Ubi (left column, white circles), LepRNull (central column, red squares) and Ubi-LepRNull (blue triangles) adult females. Abbreviations: C, prevalence of cornified/epithelial cells (proestrus or estrus); L, prevalence of leucocytes (diestrus); M, mix of leucocytes and cornified/epithelial cells (metestrus). *p<0.05; **p<0.01; ***p<0.001; ****p<0.0001 (*Figure 6— source data 1*).

DOI: https://doi.org/10.7554/eLife.40970.024

The following source data is available for figure 6:

**Source data 1.** Data regarding endocrine and reproductive changes.
DOI: https://doi.org/10.7554/eLife.40970.025

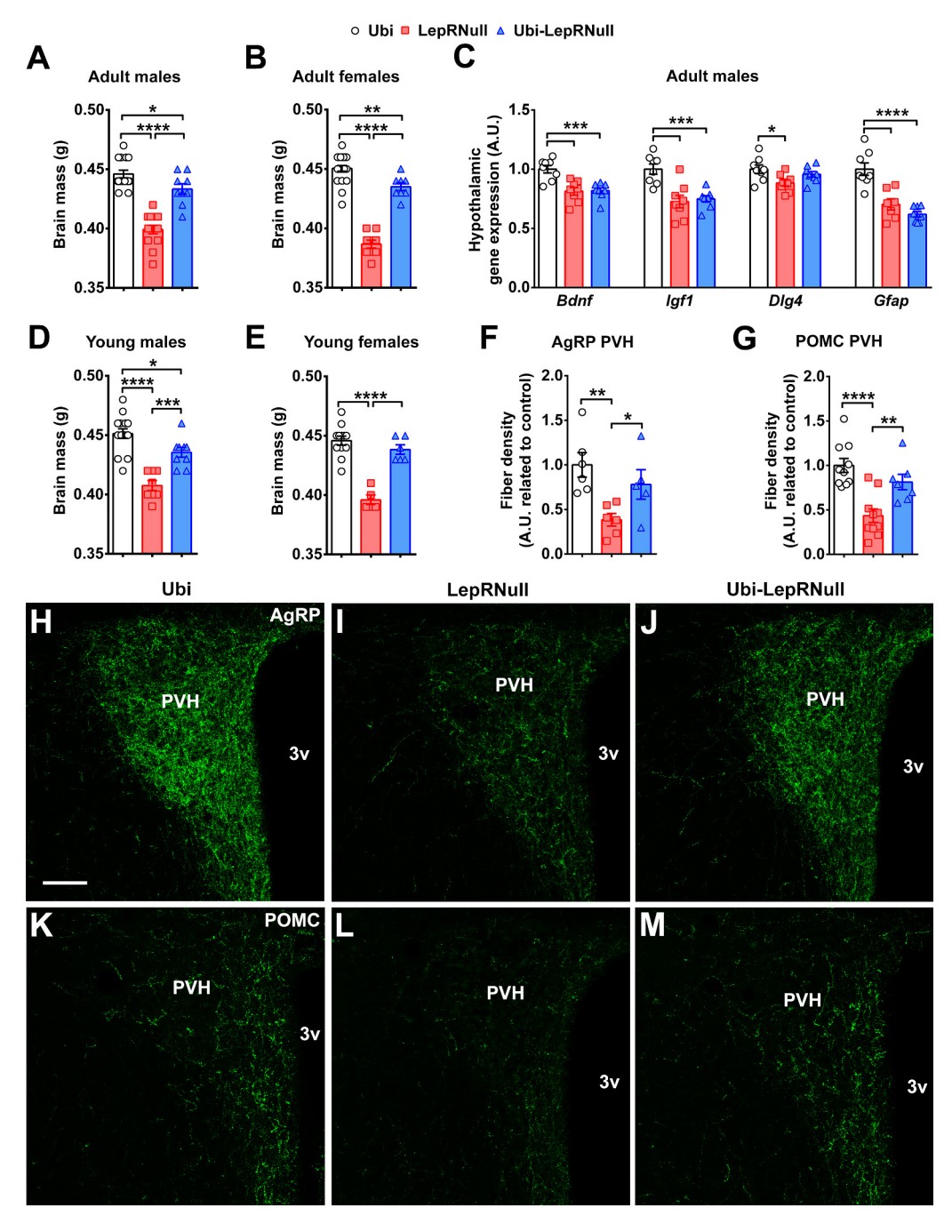

**Figure 7.** Brain development is affected by the absence of leptin signaling in early life, although ARH projections are normalized in Ubi-LepRNull mice. (A–B) Wet brain mass of male mice (*n* = 8–15) and female mice (*n* = 8–22) treated with tamoxifen at 10 weeks of age. (C) Hypothalamic mRNA expression of neurotrophic factors and proteins involved in synaptic plasticity in adult males (*n* = 7–8). (D, E) Wet brain mass of male mice (*n* = 8–15) and female mice (*n* = 5–12) treated with tamoxifen at 4 weeks of age. (F, G) Quantification of AgRP (*n* = 5–6) and POMC (*n* = 7–10) immunoreactive fibers in the PVH of adult mice. 3 v, third ventricle. Scale Bar = 50 μm. (H–J) Confocal photomicrographs of AgRP immunoreactive fibers in the PVH of Ubi (H), LepRNull (I) and Ubi-LepRNull (J) adult mice. (K–M) Confocal photomicrographs of POMC immunoreactive fibers in the PVH of Ubi (K), LepRNull (L) and Ubi-LepRNull (M) adult mice. *p<0.05; **p<0.01; ***p<0.001; ****p<0.0001 (*Figure 7—source data 1*).
DOI: https://doi.org/10.7554/eLife.40970.026

The following source data and figure supplements are available for figure 7:

*Figure 7 continued on next page*

*Figure 7 continued*

**Source data 1.** Data regarding neurotrophic effects of leptin.
DOI: https://doi.org/10.7554/eLife.40970.031
**Figure supplement 1.** No behavioral or memory deficits were found in mice that grew without leptin signaling until adulthood.
DOI: https://doi.org/10.7554/eLife.40970.027
**Figure supplement 1—source data 1.** Data regarding behavioral experiments.
DOI: https://doi.org/10.7554/eLife.40970.028
**Figure supplement 2.** *Lep^{ob/ob}* mice have a significant reduction in the amount of axons from LepR-expressing neurons in the paraventricular nucleus of the hypothalamus (PVH).
DOI: https://doi.org/10.7554/eLife.40970.029
**Figure supplement 2—source data 1.** Data regarding distribution of leptin receptor-expressing axons in the paraventricular nucleus.
DOI: https://doi.org/10.7554/eLife.40970.030

animals (*Figure 7—figure supplement 2*). This result indicates that the absence of leptin signaling reduces the density of axons from LepR-expressing neurons that innervate the PVH. Next, we evaluated the density of AgRP and POMC fibers in the PVH to determine whether LepR reactivation in adults can restore these neural projections. We confirmed that LepRNull mice had reduced density of AgRP and POMC fibers in the PVH, compared to Ubi mice (*Figure 7F–M*). However, in contrast to the findings of previous studies (*Bouret et al., 2004b*; *Kamitakahara et al., 2018*), our results indicate that restoration of leptin signaling in adults recovers the density of AgRP and POMC fibers in the PVH (*Figure 7F–M*).

## Discussion

Previous studies have shown evidence that defects in leptin signaling in early life can produce permanent consequences in brain development and energy homeostasis (*Ahima et al., 1999*; *Bouret et al., 2004b*; *Vickers et al., 2005*; *Attig et al., 2008*; *Kamitakahara et al., 2018*). On the other hand, the capacity of leptin to restore the dysfunctions of adult humans and mice previously deficient of this hormone could indicate that the action of leptin during development is not quite necessary, otherwise important sequelae should be observed despite leptin replacement (*Pelleymounter et al., 1995*; *Ioffe et al., 1998*; *Farooqi et al., 2002*; *Licinio et al., 2004*; *Donato et al., 2011*). Therefore, the exact importance of leptin signaling during development for the long-term homeostasis has not yet been fully revealed. To uncover the role of leptin during development, we studied a mouse model that grew without leptin's effects until the fourth or tenth week of life. After LepR reactivation, Ubi-LepRNull mice completely recovered their capacity to respond to leptin by exhibiting the well-known anorexigenic effect of leptin and presenting STAT3 phosphorylation in key hypothalamic nuclei after an acute leptin injection. These results are in contrast to the leptin resistance produced by early postnatal infusion of a leptin antagonist (*Attig et al., 2008*). However, adult *Lep^{ob/ob}* mice show no indication of leptin resistance since acute or chronic leptin administration produces similar responses compared to lean animals (*Halaas et al., 1995*; *Pelleymounter et al., 1995*; *Donato et al., 2011*). Thus, our findings support the idea that the lack of leptin signaling during early life does not cause leptin resistance per se in adulthood.

Despite the functional evidence that leptin responsiveness was restored in Ubi-LepRNull mice, we cannot guarantee that leptin signaling was restored in all cells that normally express LepR. Since an incomplete reactivation of LepR could lead to some defects observed in Ubi-LepRNull mice, this could be a potential issue in our model. In mice that underwent tamoxifen treatment at 10 weeks of age, we observed a tendency towards a lower hypothalamic *Lepr-b* mRNA expression compared to Ubi animals. However, it is important to mention that Ubi-LepRNull mice were hyperleptinemic, compared to lean controls, and several studies have shown that chronic hyperleptinemia can downregulate *Lepr* expression in the hypothalamus (*Martin et al., 2000*; *Ladyman and Grattan, 2005*; *Zhai et al., 2018*). In accordance with this hypothesis, *Lepr-b* mRNA expression was completely restored in mice that underwent tamoxifen treatment at 4 weeks of age and did not become

hyperleptinemic later in life, suggesting that the trend toward a lower *Lepr-b* mRNA expression in adult Ubi-LepRNull mice might be due to hyperleptinemia and not to insufficient reactivation of LepR.

Even though leptin responsiveness was recovered, Ubi-LepRNull mice still maintained a higher body weight, adiposity and serum leptin levels, compared to lean littermates. The increased body adiposity of Ubi-LepRNull mice was associated with their incapacity to normalize energy expenditure. Of note, weight loss increases energy efficiency and reduces energy expenditure beyond what is expected for the body mass loss (*Leibel et al., 1995*). Thus, to rule out a possible effect of weight loss in the metabolic dysfunctions exhibited by Ubi-LepRNull mice, we also induced the LepR reactivation in young animals, before they became obese. Since a higher body adiposity and suppressed energy expenditure were still present in those mice, our findings provide strong evidence that leptin signaling in early life is indeed required for the long-term energy homeostasis.

To further evaluate the energy homeostasis, the animals were exposed to opposing conditions of negative- and positive-energy balance. During fasting, both LepRNull and Ubi-LepRNull mice exhibited reduced drop in energy expenditure and lower rate of body weight loss. These results suggest a decreased capacity to lose weight in mice that grew without leptin signaling until adulthood. Thus, our findings are in accordance with the metabolic programming (*Samuelsson et al., 2008*), in which altered leptin signaling in early life may predispose adult individuals to metabolic diseases, such as obesity. LepRNull and Ubi-LepRNull mice also showed higher HFD intake, compared to lean controls. Since LepR reactivation completely normalized the intake of a low-fat diet, this observation indicates that the absence of leptin signaling in early life favors the consumption of a highly palatable/caloric diets. In fact, previous studies have shown that brain leptin signaling modulates the sensitivity to highly palatable food (*Hommel et al., 2006*). Therefore, our findings provide additional evidence about this effect, suggesting that disruption of leptin signaling during early life is sufficient to change the response in adulthood to palatable diets.

To uncover possible mechanisms by which the absence of leptin signaling during development caused long-term metabolic consequences, we evaluated the expression of mRNAs for key neuropeptides that regulate the energy balance. While the hypothalamic expression of orexigenic peptides, such as AgRP and NPY, were recovered in Ubi-LepRNull mice, LepR reactivation was unable to restore the mRNA levels of *Pomc*, *Cartpt* and *Prlh*. Of note, leptin signaling in POMC/CART neurons or in hypothalamic Prlh cells influences the energy balance predominantly via changes in energy expenditure and not through food intake (*Berglund et al., 2012*; *Dodd et al., 2014*). Therefore, the mild obesity and lower metabolic rate of Ubi-LepRNull mice are likely associated with defects in POMC or Prlh neurons. Interestingly, previous studies have shown that the absence of hypothalamic POMC expression in early life also causes long-term metabolic imbalances, even though POMC expression is rescued in adulthood (*Bumaschny et al., 2012*; *Chhabra et al., 2016*). However, differently than Ubi-LepRNull mice, early POMC deficiency leads to increased food intake later in life. In addition, if POMC reactivation was induced in non-obese animals or at the fourth week of life, no metabolic imbalances were observed in adulthood (*Bumaschny et al., 2012*; *Chhabra et al., 2016*). In the case of LepR reactivation on the fourth week of life, we still observed a higher body adiposity and lower energy expenditure in Ubi-LepRNull mice, compared to lean controls.

HDAC proteins can produce epigenetic modifications via histone deacetylation and chromatin remodeling (*Haberland et al., 2009*). In addition, recent evidence indicates that HDAC5 regulates energy homeostasis via hypothalamic neurons (*Kabra et al., 2016*). Importantly, HDAC5 induces site-specific acetylation changes in POMC neurons and knockdown of hypothalamic *Hdac5* suppresses *Pomc* mRNA levels in adult rats or mice, without affecting *Agrp* expression (*Kabra et al., 2016*). Since hypothalamic expression of *Hdac3* and *Hdac5* were not restored after LepR reactivation, histone-mediated epigenetic modifications in POMC neurons could be involved with the energy imbalance caused by the absence of leptin signaling in early life. The lack of leptin signaling during development may also have produced epigenetic changes via DNA methyltransferases, since hypothalamic *Dnmt3a* and *Dnmt3b* expression remained suppressed in Ubi-LepRNull mice. Accordingly, *Dnmt3a* ablation in PVH neurons causes obesity and decreased energy expenditure in mice (*Kohno et al., 2014*). However, whether epigenetic changes were indeed responsible for the metabolic dysfunctions of Ubi-LepRNull mice will require additional studies.

Previous reports have shown that metabolic imbalances during development alter insulin signaling in adulthood (*Chen et al., 2009*; *Vogt et al., 2014*). We observed that LepR reactivation in adult

animals caused insulin resistance, without impairing glucose tolerance of Ubi-LepRNull mice. Since obesity leads to pancreatic beta cell hyperplasia (*Bock et al., 2003*), the normal glucose tolerance of Ubi-LepRNull mice could be explained by an obesity-induced increase in insulin secretory capacity, overcoming their insulin resistance. In contrast, LepR reactivation before the onset of obesity possibly prevented major pancreatic islet changes. Therefore, glucose homeostasis in our experimental animals was predominantly affected by the former obesity rather than by the absence of leptin signaling during development.

Leptin deficiency causes hypogonadotropic hypogonadism, low testicle weight, anestrus and infertility (*Chehab et al., 1996*; *Mounzih et al., 1997*; *Farooqi et al., 2002*; *Licinio et al., 2004*). LepR reactivation in adult animals was not sufficient to completely recover the reproductive system in both genders. However, leptin treatment rescues the sterility of male and female *Lep^{ob/ob}* mice (*Chehab et al., 1996*; *Mounzih et al., 1997*; *Donato et al., 2011*). The reasons of these contrasting results are unknown, but may be related to genetic differences between *Lep^{ob/ob}* and LepRNull mice or to the age at which the animals were studied. In the present study, the reproductive system was evaluated in 6- to 8-month-old animals, which is an older age to assess fertility than usual. This advanced age may explain why only half of females breeding with Ubi mice got pregnant. The alterations in reproduction observed in Ubi-LepRNull mice probably had a central cause since reductions in *Kiss1* and *Nos1* mRNA levels were observed in their hypothalami. These transcripts produce neurotransmitters that stimulate GnRH release (*Messager et al., 2005*; *Bellefontaine et al., 2014*). As a result, hypothalamic GnRH fiber density was not different between LepRNull and Ubi-LepRNull mice, indicating defects in the capacity to release GnRH in the hypophyseal portal system (*Polkowska et al., 2006*; *Donato et al., 2011*). Thus, our findings suggest that the absence of leptin signaling during development causes long-term defects in the neurocircuits that regulate the hypothalamic-pituitary-gonadal axis.

In adult animals, leptin responsive neurons are mostly confined to hypothalamic and brainstem nuclei involved in feeding and neuroendocrine regulation (*Ramos-Lobo and Donato, 2017*). However, during early postnatal life, *Lepr* mRNA expression is also detected in other brain regions, including the cerebral cortex, hippocampus and thalamus, suggesting a potential role of leptin in brain development (*Caron et al., 2010*). In this sense, we confirmed the results of previous studies indicating that leptin deficiency leads to reduced brain mass (*Ahima et al., 1999*). Interestingly, we also observed a sexually dimorphic response when LepR reactivation occurred in 4-week-old mice. In this case, Ubi-LepRNull females completely recovered their brain mass, whereas Ubi-LepRNull males exhibited a partial increase. *Ahima et al., 1999* studied only female *Lep^{ob/ob}* mice and showed a greater recovery in brain mass when leptin replacement started at 4 weeks of life, compared with a later treatment. Therefore, males and females may present distinct critical postnatal periods, in which leptin acts in order to allow a normal brain development. The remarkable increase in brain mass after LepR reactivation can be explained by an increase in cell number and/or edema. Previous studies have shown conflicting results regarding the causes of the reduced brain mass in *Lep^{ob/ob}* mice. While some authors found reduced DNA content in the brain of *Lep^{ob/ob}* mice, indicating lower number of cells (*van der Kroon and Speijers, 1979*), other studies showed decreased soma cross-sectional area in neurons of *Lep^{ob/ob}* mice compared to controls (*Bereiter and Jeanrenaud, 1979*). Furthermore, *Lep^{ob/ob}* and wild-type mice exhibit similar brain protein content (*Sena et al., 1985*). Therefore, additional experiments are still necessary to investigate the possible mechanisms behind the increase in brain mass after LepR reactivation. Despite the reduction in brain mass, no evidence of behavioral or memory deficit was found in mice that grew without leptin signaling until adulthood. In humans, leptin replacement in a 5-year-old leptin-deficient boy rescued the cognitive development which was slower than expected for his age (*Paz-Filho et al., 2008*).

Our results are in accordance with earlier studies indicating that the absence of leptin signaling disrupts the normal developmental pattern of projections from the ARH to the PVH (*Bouret et al., 2004b*; *Bouret et al., 2004a*). However, an unexpected finding was the recovery of the density of AgRP and POMC fibers in the PVH of Ubi-LepRNull mice, even though the restoration of leptin signaling was induced several weeks after the previously indicated critical period for the trophic action of leptin (*Bouret et al., 2004b*; *Kamitakahara et al., 2018*). In our study, these projections were investigated by staining AgRP or POMC-derived peptides, which not necessarily allow the complete visualization of ARH neuron axons. So, it is uncertain whether LepR reactivation promoted axon extension into the PVH or just improved the visualization of processes that were already there.

However, *Bouret et al. (2004b)* used the anterograde tracer DiI to demonstrate that this neurocircuit does not fully develop without leptin signaling. Additionally, we found a significant reduction in the amount of axons from LepR-expressing neurons in the PVH of *Lep^{ob/ob}* mice compared to control animals. Thus, these findings suggest that the higher density of AgRP and POMC fibers in the PVH after LepR restoration was likely caused by axonal growth.

The reasons for the divergent findings between our study and previous reports that indicated a critical postnatal period for the trophic action of leptin (*Bouret et al., 2004b*; *Kamitakahara et al., 2018*) are unknown, but may be related to methodological differences. While previous studies investigated the neurotrophic actions of leptin via repeated injections in *Lep^{ob/ob}* mice (*Bouret et al., 2004b*; *Kamitakahara et al., 2018*), our study allowed the endogenously produced leptin to act in a physiological manner. Thus, the circadian rhythm of serum leptin was present in our mouse model and large variations in leptin concentrations, common in acute injections, were avoided (*Ahima et al., 1996*; *Ahima et al., 1998*). Furthermore, chronic injections can produce stress in rodents and are not feasible in the long-term, representing confounders that could have affected leptin's trophic effect in earlier studies (*Bouret et al., 2004b*; *Kamitakahara et al., 2018*). Nevertheless, our results indicating that ARH projections can be recovered even in adult animals are consistent with the fact that exogenous leptin treatment is able to induce a robust weight loss and improvements in a variety of neuroendocrine and metabolic dysfunctions of *Lep^{ob/ob}* mice (*Maffei et al., 1995*; *Montague et al., 1997*; *Farooqi et al., 2002*; *Bellefontaine et al., 2014*).

### Concluding remarks

Our findings revealed that the absence of leptin signaling in early life led to permanent changes in energy homeostasis, melanocortin system, reproduction and brain development. The metabolic dysfunctions caused by the lack of leptin signaling during development possibly involved changes in neural populations that regulate the energy expenditure, such as POMC or Prlh neurons in the hypothalamus, as well as epigenetic mechanisms since the hypothalamic expression of several enzymes that modulate DNA methylation or histone acetylation were affected in Ubi-LepRNull mice. Although the complete absence of leptin signaling is rarely found in humans, both undernutrition and overnutrition are able to produce significant changes in serum leptin concentrations, leptin sensitivity or in the development of neurocircuits that regulate energy homeostasis (*Samuelsson et al., 2008*; *Chen et al., 2009*; *Vogt et al., 2014*; *Ralevski and Horvath, 2015*). Therefore, intrauterine or early postnatal changes in energy balance have been consistently linked with obesity and other metabolic diseases in adulthood (*Vickers et al., 2005*; *Attig et al., 2008*; *Lillycrop and Burdge, 2011*; *Bumaschny et al., 2012*; *Chhabra et al., 2016*). Thus, our findings contribute to the understanding of the metabolic programming by indicating which specific metabolic, neuroendocrine and developmental consequences arise from the absence of leptin signaling in early life, especially in the context of the alarming growth of childhood obesity worldwide (*Cunningham et al., 2014*).

## Materials and methods

### Key resources table

| Reagent type (species) or resource | Designation | Source or reference | Identifiers | Additional information |
|---|---|---|---|---|
| Genetic reagent (*M. musculus*) | STOCK Lepr^{tm1Jke}/J | The Jackson Laboratory | JAX:018989 | |
| Genetic reagent (*M. musculus*) | B6.Cg-*Ndor 1^{Tg(UBC-cre/ERT2)1Ejb}*/2J | The Jackson Laboratory | JAX:008085 | |
| Genetic reagent (*M. musculus*) | B6.Cg-*Lep^{ob}*/J | The Jackson Laboratory | JAX:000632 | |
| Genetic reagent (*M. musculus*) | B6.129-Lepr^{tm2(cre)Rck}/J | The Jackson Laboratory | JAX:008320 | |
| Genetic reagent (*M. musculus*) | B6;129S6-Gt(ROSA) 26Sor^{tm9(CAG-tdTomato)Hze}/J | The Jackson Laboratory | JAX:007909 | |

*Continued on next page*

*Continued*

| Reagent type (species) or resource | Designation | Source or reference | Identifiers | Additional information |
|---|---|---|---|---|
| Antibody | Rabbit anti-GnRH (LHRH) antibody | Immunostar | RRID:AB_572248 | IF (1:2000) |
| Antibody | Rabbit anti-pSTAT3 Tyr705 antibody | Cell Signaling | Cat. #: 9131 | IHC (1:1000) |
| Antibody | Mouse anti-αMSH antibody | Chemicon | Cat. #: AB5087 | IF (1:2000) |
| Antibody | Rabbit anti-βendorphin antibody | Phoenix Pharmaceuticals | Cat. #: H-022–33 | IF (1:2000) |
| Antibody | Rabbit anti-AgRP antibody | Phoenix Pharmaceuticals | Cat. #: H-003–53 | IF (1:2000) |
| Antibody | Biotin-SP-conjugated AffiniPure Donkey anti-Rabbit IgG | Jackson ImmunoResearch | Cat. #: 711-065-152 | IHC (1:1000) |
| Antibody | Alexa Fluor488-conjugated Donkey anti-Rabbit IgG | Jackson Immuno Research | Cat. #: 711-545-152 | IF (1:500) |
| Antibody | Alexa Fluor594-conjugated Donkey anti-Mouse IgG | Jackson Immuno Research | Cat. #: 715-585-150 | IF (1:500) |
| Sequence-based reagent | RT-qPCR primers | This paper | | |
| Peptide, recombinant protein | Human recombinant insulin | Novo Nordisk | | |
| Peptide, recombinant protein | Mouse recombinat leptin | National Hormone and Peptide Program | | |
| Commercial assay or kit | Mouse Leptin ELISA KIT | Crystal Chem | 90030 | |
| Commercial assay or kit | Total Rat/Mouse T4 ELISA KIT | Calbiotech | T4044T-100 | |
| Commercial assay or kit | Rat/Mouse Testosterone ELISA KIT | Calbiotech | TE187S-100 | |
| Commercial assay or kit | Mouse Insulin ELISA KIT | Crystal Chem | 90080 | |
| Commercial assay or kit | Corticosterone EIA KIT | Arbor Assays | K014-H1 | |
| Chemical compound, drug | Tamoxifen | Sigma-Aldrich | T5648 | |
| Chemical compound, drug | Sesame oil | Sigma-Aldrich | S3547 | |
| Software, algorithm | ImageJ | National Institutes of Health (NIH) | http://rsb.info.nih.gov/ij/ | |
| Software, algorithm | Prism | GraphPad | https://www.graphpad.com/scientific-software/prism/ | Version 6 |
| Software, algorithm | ANY-maze | ANY-maze | http://anymaze.co.uk/ | |
| Other | High-fat diet (HFD) 5.31 kcal/g, 58% calories from fat | Pragsoluções | | |

## Mice

Experiments were performed in male and female mice. To induce reactivation of the *Lepr* gene, a *Lox*-flanked transcription-blocking cassette (*lox*TB) was inserted in the *Lepr* gene to generate mice null for the *Lepr* (STOCK *Lepr*^tm1Jke/J, The Jackson Laboratory). They were bred with animals expressing Cre-ERT2 fusion protein under the human ubiquitin C promoter sequence (B6.Cg-*Ndor1*^Tg(UBC-cre/ERT2)1Ejb/2J, The Jackson Laboratory). As control animals, we used obese mice homozygous for the *lox*TB allele (LepRNull group) and lean control mice carrying only the Cre allele (Ubi

group). Mice carrying both the Cre transgene and the loxTB allele in homozygosity were considered the conditional knockout mice (Ubi-LepRNull group). Mice in these strains were in the C57BL/6 background and all groups were composed of littermate animals. To visualize LepR-expressing cells, we generated a LepR-reporter mouse by breeding the LepR-Cre mouse (B6.129-Lepr[tm2(cre)Rck]/J, The Jackson Laboratory) with the lox-stop-lox(LSL)-tdTomato reporter mouse (B6;129S6-Gt(ROSA) 26Sor[tm9(CAG-tdTomato)Hze]/J, The Jackson Laboratory). Thus, axons from LepR-expressing cells were visualized without additional staining by the presence of the tdTomato fluorescent protein. We then crossed Lep[ob/+] mice (B6.Cg-Lep[ob]/J, The Jackson Laboratory) with LepR-Cre::LSL-tdTomato mice, in order to produce leptin deficient mice expressing the tdTomato fluorescent protein only in LepR cells. Mice were weaned at 3–4 weeks of age and their mutations were confirmed by genotyping the DNA that had been previously extracted from the tail tip (REDExtract-N-Amp Tissue PCR Kit, Sigma). The genetically modified mouse models were produced and maintained in standard conditions of light (12 hr light/dark cycle) and temperature (22 ± 1°C). Unless otherwise indicated, mice received a regular rodent low-fat chow diet (2.99 kcal/g; 9.4% calories from fat). All experiments were carried out in compliance with NIH guidelines for the care and use of laboratory animals and were previously approved by our Institutional Animal Ethics Committee (protocol number 137/2013).

## Temporal reactivation of the LepR

All mice received 5 i.p. injections of tamoxifen (0.15 mg/g, Sigma-Aldrich), diluted in sesame oil (Sigma-Aldrich), in alternate days. Mice were treated with tamoxifen at 10 weeks of age (adults; LepRNull and Ubi-LepRNull mice were already obese) or at 4 weeks of age (young; before the onset of obesity). The animals were monitored until stabilization of body weight and food intake. After the stabilization period, the in vivo experiments were carried out.

## Evaluation of energy and glucose homeostasis

At least 6 weeks after the last tamoxifen injection (16 weeks of age in adults or 15 weeks of age for young mice), mice were single housed. Their body weight and food intake were measured for 3 to 4 alternate days. Then, mice were subjected to a glucose tolerance test (2 g glucose/kg b.w.; i.p.) and to an insulin tolerance test (1 IU insulin/kg b.w.; i.p.). To determine $O_2$ consumption (energy expenditure), $CO_2$ production, respiratory exchange ratio (RER) and locomotor activity (through infrared beam sensors) mice were placed in the Oxymax/Comprehensive Lab Animal Monitoring System (CLAMS; Columbus Instruments, Columbus, OH). After an adaptation period of 3 days inside the CLAMS, these metabolic parameters were evaluated for four consecutive days. Therefore, the results presented are the average of this period. Body adiposity was determined by weighting perigonadal (PG), subcutaneous (SC) and retroperitoneal (RP) fat pads in males. For the females, the periovarian (Ov) fat pad was also included.

## Brain histology

To visualize leptin responsive cells in the brain, adult mice (n = 3–4/group) received an acute i.p. injection of mouse recombinant leptin (5 µg/g, from Dr. A.F. Parlow, National Hormone and Peptide Program – NHPP, National Institute of Diabetes and Digestive and Kidney Diseases) and were perfused 90 min later. For the perfusions, mice were deeply anesthetized with isoflurane and perfused transcardially with saline, followed by a 10% buffered formalin solution (150–200 mL per mouse). Brains were collected and post-fixed in the same fixative for 60 min and cryoprotected overnight at 4°C in 0.1 M PBS containing 20% sucrose, pH 7.4. Brains were cut (30 µm thick sections) in the frontal plane using a freezing microtome. To label pSTAT3, brain sections were rinsed in 0.02 M potassium PBS, pH 7.4 (KPBS), followed by pretreatment in an alkaline (pH >13) water solution containing 1% hydrogen peroxide and 1% sodium hydroxide for 20 min. After rinsing in KPBS, sections were incubated in 0.3% glycine and 0.03% lauryl sulfate for 10 min each. Next, sections were blocked in 3% normal donkey serum for 1 hr, followed by incubation in anti-pSTAT3[Tyr705] primary antibody (1:1000; Cell Signaling; #9131) for 40 hr. AgRP, β-endorphin, α-MSH or GnRH immunoreactivity was evaluated in brain sections that were rinsed in KPBS and blocked in 3% normal donkey serum for 1 hr, followed incubation in anti-AgRP (1:2000, Phoenix Pharmaceuticals; #H-003–53), anti-β-endorphin (1:2000, Phoenix Pharmaceuticals; #H-022–33), anti-αMSH (1:2000, Chemicon; #AB5087) or anti-GnRH (1:2000, Immunostar; #AB_572248) primary antibodies. For the immunofluorescence reaction,

sections were rinsed in KPBS and incubated for 90 min in AlexaFluor[488] or AlexaFluor[594]-conjugated secondary antibodies (1:500, Jackson Laboratories). Sections were mounted onto gelatin-coated slides and the slides were coverslipped with Fluoromount G (Electron Microscopic Sciences, Hatfield, PA). For the immunoperoxidase staining, sections were incubated for 1 hr in biotin-conjugated secondary antibody (1:1000, Jackson Laboratories) and followed by 1 hr incubation with an avidin-biotin complex (1:500, Vector Labs). The peroxidase reaction was performed using 0.05% 3,3'-diaminobenzidine, 0.25% nickel sulfate and 0.03% hydrogen peroxide resulting in a black nuclear staining. The slides were coverslipped with DPX mounting medium (Sigma, St. Louis, MO). Brightfield and epifluorescence photomicrographs were acquired with a Zeiss Axiocam HRc camera coupled to a Zeiss Axioimager A1 microscope (Zeiss, Munich, Germany). Images were digitized using Axiovision software (Zeiss). Confocal images were captured in a 1024 × 1024 pixel format using a Zeiss LSM 780 confocal laser scanning inverted microscope (Carl Zeiss, Germany). Image stacks comprised eight images captured with a LD Plan-Neofluar 40x/0.6 Korr M27 objective (Zeiss). The ImageJ Cell Counter software (http://rsb.info.nih.gov/ij/) was used to manually count the number of cells in the areas of interest. Fiber density was determined by measuring the integrated optic density using ImageJ software.

## Leptin responsiveness

To assess leptin sensitivity, adult mice received an i.p. injection of either phosphate-buffered saline (PBS) or mouse recombinant leptin (2.5 µg/g b.w.; from Dr. A.F. Parlow, NHPP, USA) 3 hr before their dark phase, and their food intake and body weight were recorded for 24 hr following the injection. Both parameters after PBS injection were compared with the values after leptin administration.

## Metabolic effects induced by fasting or acute HFD exposure

To investigate the response to metabolic challenges of negative or positive energy balance, previously single-housed mice were initially subjected to a 24 hr fasting protocol. During this challenge, metabolic parameters were continuously assessed in the CLAMS, and their body weight was monitored at baseline and after 24 hr of fasting. Changes in metabolic parameters during fasting were reported as percentage of the values obtained from baseline (fed state). Food intake was recorded 4, 12, 24 and 48 hr following the fasting period. After a recovery period (at least 1 week), the animals were exposed to a HFD (5.31 kcal/g, 58% calories from fat) for 48 hr. Metabolic parameters were assessed in the CLAMS, together with their body weight and food intake (normalized by calorie consumption).

## Changes in the reproductive system

To investigate changes in the reproductive system, we weighted the testicles of adult and young males. To assess fertility in males, Ubi and Ubi-LepRNull adult males (n = 5–6/group) were mated with C57BL6 female mice (two females for each male) for 28 days and the number of pregnant females from each group was compared. In females, we monitored the estrous cycle daily for 25 days (n = 6/group). The proportion of cell types in the vaginal smear was determined and the cycle phases were classified as follows: diestrus (mostly leucocytes), metestrus (mixture of leucocytes and cornified cells) and proestrus/estrus (mostly epithelial or cornified cells).

## Hormone measurements

Commercially available ELISA kits were used to determine the serum concentration of leptin (Crystal Chem), T4 (Calbiotech), testosterone (Calbiotech), insulin (Crystal Chem), and corticosterone (Arbor Assays).

## Gene expression analysis

Total RNA from the hypothalamus was extracted with TRIzol reagent (Invitrogen). Assessment of RNA quantity and quality was performed with an Epoch Microplate Spectrophotometer (Biotek). Total RNA was incubated with DNase I RNase-free (Roche Applied Science). Reverse transcription was performed with 2 µg of total RNA with SuperScript II Reverse Transcriptase (Invitrogen) and random primers p(dN)6 (Roche Applied Science). Real-time polymerase chain reaction was performed using the 7500TM Real-Time PCR System (Applied Biosystems) and Power SYBR Green PCR Master

Mix (Applied Biosystems). Relative quantification of mRNA was calculated by $2^{-\Delta\Delta Ct}$. Data were normalized to the geometric average of *Actb*, *Gapdh* and *Ppia* and reported as fold changes compared to values obtained from the control group (set at 1.0). The list of primers is available as *Figure 3—source data 2*.

### Brain and behavioral modifications

The wet brain mass was determined in adult and young mice, both males and females. All behavioral tests were performed in male mice treated with tamoxifen at 10 weeks of age. For the open-field test, mice were placed in an open field arena (0.3 m (w) x 0.3 m (d) x 0.45 m (h) for 5 min and monitored by the ANY-maze software, which tracks animal's position and records the distance travelled. To investigate spatial learning/memory, we performed the Barnes maze test. The maze consists of a white circular platform (92 cm of diameter) elevated 105 cm above the floor with 20 equally spaced holes (5 cm diameter and 7.5 cm between holes) along the perimeter with a darkened escape box placed in a constant position under one of the holes at the edge of the platform. Mice are presumed to learn the location of an escape hole using extra-maze visual cues placed on the walls of the room. Mice were initially subjected to one habituation phase in the first day prior to the initial trial. This phase was performed by placing the animal in a cylindrical chamber in the middle of the maze for 10 s. Afterwards, the chamber was removed, and mice were gently guided to the escape box. Animals were allowed to remain in the escape box for 2 min and then were returned to their original cage. The acquisition phase consisted of four trials per day for four days. For each trial, mice were placed in the cylindrical chamber in the middle of the maze for 10 s and then they could explore the maze for 3 min. After trial's completion (i.e mice either entered the escape box or remained exploring the maze for 3 min) animals were gently guided to and remained inside the escape box for 1 min. Subsequently, mice were returned to their home cages until the next trial, with a 15 min interval between trials. Time (primary latency) and number of errors (primary errors) to reach the escape hole for the first time were recorded by a trained researcher. Errors were considered as the number of head deflections into incorrect holes of the maze and were counted by the experimenter. Test sessions were performed 24 hr and 1 week after the last trial (days 5 and 12, respectively) and consisted of 90 s exploring the maze with the escape box closed. In all experimental steps, the maze was cleaned with 40% ethanol and air-dried before each trial and between mice. The platform was monitored by the ANY-maze software, which recorded the animal's position and the distance the animal moved (total distance travelled).

### Statistics

All results were expressed as mean ±s.e.m. We performed Bartlett's test of normality to evaluate for Gaussian distribution of samples. When data had normal distribution, we used one-way ANOVA and the Newman-Keuls multiple comparison post-hoc test. In cases of non-parametric data, we used Kruskal-Wallis followed by Dunn's test. In specific cases, we used the Mann-Whitney test with Bonferroni's correction. Data was analyzed using two-way ANOVA when appropriate, followed by Newman-Keuls post-hoc test. Statistical analyses were performed using GraphPad Prism software. We considered p values < 0.05 to be statistically significant and p<0.01 when Bonferroni's correction was applied.

## Acknowledgements

We thank Ana Maria P Campos and André M Lima for the technical assistance, and Dr. Joel K Elmquist (University of Texas Southwestern Medical Center, Dallas, TX) for kindly providing the LepRNull mouse. JD was financed by grants from the São Paulo Research Foundation (FAPESP; 15/10992–6). AM.RL and ICF were financed by fellowships from FAPESP (14/11752–6 and 16/09679–4, respectively).

# Additional information

## Funding

| Funder | Grant reference number | Author |
|---|---|---|
| Fundação de Amparo à Pesquisa do Estado de São Paulo | 15/10992-6 | Jose Donato Jr |
| Fundação de Amparo à Pesquisa do Estado de São Paulo | 14/11752-6 | Angela M Ramos-Lobo |
| Fundação de Amparo à Pesquisa do Estado de São Paulo | 16/09679-4 | Isadora C Furigo |

The funders had no role in study design, data collection and interpretation, or the decision to submit the work for publication.

## Author contributions

Angela M Ramos-Lobo, Data curation, Formal analysis, Investigation, Writing—review and editing; Pryscila DS Teixeira, Isadora C Furigo, Investigation, Writing—review and editing; Helen M Melo, Natalia de M Lyra e Silva, Formal analysis, Investigation, Methodology, Writing—review and editing; Fernanda G De Felice, Formal analysis, Supervision, Methodology, Writing—review and editing; Jose Donato Jr, Conceptualization, Resources, Formal analysis, Supervision, Funding acquisition, Investigation, Writing—original draft, Project administration

## Author ORCIDs

Jose Donato Jr (iD) http://orcid.org/0000-0002-4166-7608

## Ethics

Animal experimentation: All experiments were carried out in compliance with NIH guidelines for the care and use of laboratory animals and were previously approved by our Institutional Animal Ethics Committee (protocol number 137/2013).

## Decision letter and Author response

Decision letter https://doi.org/10.7554/eLife.40970.034
Author response https://doi.org/10.7554/eLife.40970.035

# Additional files

## Supplementary files
• Transparent reporting form
DOI: https://doi.org/10.7554/eLife.40970.032

## Data availability

Individual values were plotted in each figure and source data files have also been included.

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
