## [Decision Letter]

Thank you for submitting your article "Long-term consequences of the absence of leptin signaling in early life" for consideration by *eLife*. Your article has been reviewed by three peer reviewers, including Richard D Palmiter as the Reviewing Editor and Reviewer #1, and the evaluation has been overseen by Catherine Dulac as the Senior Editor.

The reviewers have discussed the reviews with one another and the Reviewing Editor has drafted this decision to help you prepare a revised submission.

Summary:

The authors of this paper demonstrate that restoration of leptin signaling at 4 or 10 weeks of age in mice that previously had no leptin signaling is able to restore all metabolic phenotypes and partially restore fertility and neurological phenotypes. These experiments suggest that there is not an absolute critical period early in development when leptin signaling is essential.

Essential revisions:

There was general agreement among the reviewers that this paper makes an important contribution and should be published after adequate revision. Some issues that the authors should address summarized here and the original reviews are also included below to help delineate the concerns. There are two possibilities to explain lack of full restoration of any parameter in this type of study: (a) incomplete restoration of LepR expression in critical cells, (b) the existence of a critical period after which it is no longer possible to restore function, or (c) a combination of the two.

1) In the case of fertility, it has previously been shown that restoration of leptin signaling only in AgRP neurons (Anderson) is sufficient to restore fertility. Restoration of leptin signaling is thought to reduce inhibitory signaling to nearby Kiss1 neurons and thereby allow Kiss1 neurons to activate GnRH neurons. Consistent with this view, ablation of AgRP neurons in leptin-deficient mice is also able to restore fertility (Palmiter). Thus, the incomplete restoration of fertility in this paper may be due to incomplete restoration of LepR signaling specifically in AgRP neurons. The authors should quantify the degree of restoration of LepR expression in AgRP neurons.

2) The ability to partially restore brain weight in adult mice is remarkable, even if it is incomplete. Since it is not known where leptin signaling is important for regulation brain weight, a more thorough analysis of Lepr signaling in specific cells is not feasible. However, it is important to know whether the increase in brain weight is due to an increase in cell number (DNA content) or edema. The former result would suggest that future investigations into what cells proliferate in the adult and where they are located would be warranted.

3) The long-standing observation that there is a critical period in early development for AgRP and POMC neuron maturation is challenged by the current results. The data in Figure 7 reveal extensive restoration of AgRP and POMC immunofluorescence in one major target (PVN) of AgRP and POMC neurons. The question is whether restoration of LepR signaling in adult mice promoted axon extension into the PVN or whether the axons were there all along, but lacked AgRP and POMC due to inadequate synthesis or trafficking. Ideally, the authors should design an experiment to visualize the AgRP and POMC axons in the PVN independently of these peptides. Alternatively, a cogent discussion of the issue is warranted.

*Reviewer #1:*

This paper represents a comprehensive study of the consequences of restoring leptin receptor function in mice with a null mutation in that receptor gene on many aspects of physiology, anatomy and behavior. The restoration was either done at 4 weeks (before obesity) or at 10 weeks, when the mice were already quite obese. Virtually all parameters that were measured improved after restoration of the LepR, but in some cases there were still small differences compared to normal littermates. Those differences presumably represent permanent changes that cannot be fully reversed, e.g. brain weight can't be restored in the absence of neurogenesis. Fertility is one parameter that was not fully restored, whereas previous studies have suggested that it can be restored with expression of LepR only in AgRP neurons and ablation of AgRP neurons in Lep *ob/ob* mice also restores fertility. Thus, a more thorough analysis of LepR activity in AgRP neurons is warranted because Ubc-Cre may not be fully functional in those critical neurons. The current results contradict previous studies indicating that leptin signaling is required during a critical period to promote axon outgrowth (or neuropeptide accumulation) in target tissues. The results presented here reveal that neuropeptide immunostaining is fully restored even after 10 weeks without leptin signaling. Bolstering this result with additional approaches would be a valuable addition. It seems unlikely that growth of new axons (fibers) into target tissues actually occurs after restoring leptin signaling. A more likely possibility is that the fibers are there but not visible due to low peptide abundance. Distinguishing between these possibilities would be a welcome addition.

*Reviewer #2:*

The manuscript by Donato et al. describes the detailed analysis of mice in which Lepr expression has been re-established in a Cre-dependent manner at 4 and 10 weeks after birth. In large, while Lepr re-expression restores to large extend altered food intake and body weight gain and hypothalamic melanocortin projections in mice previously deficient for the Lepr, it predisposes for increased fat mass, reduced energy expenditure and impaired reproductive function. Collectively the study defines a subset of phenotypes which depend on developmental leptin action.

The study seems very carefully performed, and the results overall support the conclusions by the authors. the paper is clearly written and should be of interest to the audience of *eLife*.

However, one major concern should be addressed experimentally:

The described restoration of hypothalamic projections appears difficult to comprehend, i.e. that neuron send projections after 10 weeks of neurocircuitry development. This clearly represents a very interesting finding. Nevertheless, currently this is based on staining for endogenous peptides whose expression is altered in the models. Thus, complementary experiments with genetically marking anatomical projections independent of the endogenously expressed peptides should be performed to substantiate this finding.

*Reviewer #3:*

In their manuscript "Long-term consequences of the absence of leptin signaling in early life," Ramos-Lobo and colleagues study the effects of restoring leptin receptor expression in mice with genetic deficiency of LepR. They conclude that some aspects of LepR deficiency are due to developmental consequences of LepR-deficiency early in development. They conclude that reproductive and neuroanatomical phenotypes are especially sensitive to early leptin signaling. The data are of excellent quality. Based on their data, it is fair to state that restoration of leptin receptor signaling at 10 weeks restores gross metabolic function and hypothalamic PVH projections in mice lacking Lepr at conception. However, it is harder to conclude that lack of complete restoration of phenotypes in their model is because of developmental defects. It is also possible, as they point out, that the lack of restoration of phenotypes is due to less than complete efficiency of restoration of Lepr expression.

What the authors can conclude is that the restoration of Lepr signaling achieved at 10 weeks in their system largely corrects body weight, feeding, energy expenditure phenotypes but has much more modest effects on reproductive and neuroanatomical phenotypes. This qualitative difference in phenotype restoration may be due to a permanent developmental defect or to a greater degree of sensitivity to leptin signaling of certain phenotypes.

The authors do acknowledge this shortcoming. It might be helpful to quantify the restoration of LepR expression in populations of neurons to determine efficiency. However, there is an inherent challenge of any system in which restoration of genetic mutation is not 100% efficient and one does not achieve a complete restoration of wild-type phenotypes.

---

## [Author Response]

Essential revisions:1) In the case of fertility, it has previously been shown that restoration of leptin signaling only in AgRP neurons (Anderson) is sufficient to restore fertility. Restoration of leptin signaling is thought to reduce inhibitory signaling to nearby Kiss1 neurons and thereby allow Kiss1 neurons to activate GnRH neurons. Consistent with this view, ablation of AgRP neurons in leptin-deficient mice is also able to restore fertility (Palmiter). Thus, the incomplete restoration of fertility in this paper may be due to incomplete restoration of LepR signaling specifically in AgRP neurons. The authors should quantify the degree of restoration of LepR expression in AgRP neurons.

We would like to thank the editors and the reviewers for the valuable comments and suggestions. We understand the concern regarding the possible lack of complete reactivation of leptin receptor (LepR) as the main cause of the defects observed in our mouse model. This limitation was acknowledged in the second paragraph of the Discussion section. Following the reviewers’ suggestion, we have quantified the degree of restoration of LepR expression in AgRP neurons by co-localizing pSTAT3 in AgRP cells of leptin-injected Ubi-LepRNull mice using double-labeled immunoperoxidase reaction (Figure below). Using this approach, virtually all AgRP-labeled cells expressed leptin-induced pSTAT3 in Ubi-LepRNull mice (*n* = 2). In Author response image 1, arrows indicate high magnification photomicrographs of representative double-labeled neurons. pSTAT3 appears as a black nucleus and AgRP expression as a brownish cytoplasm.

To address the reviewers’ concern, we recognize that the best approach would be to add a AgRP-reporter gene to the background of UbiquitinC^ERT2^::LepR^Null/Null^ mice to improve this co-localization, but such approach would require many months until we produce the first Ubi-LepRNull mice carrying a reporter protein under the *Agrp* promoter, in addition to the LepR reactivation procedure, which takes several weeks. This long timeline prevented us from carrying out this approach. Instead, we performed the double-labeled immunoperoxidase approach even though we recognize that the co-localization result presented is not ideal since the soma of AgRP cells is poorly stained (AgRP peptide is mainly present in axons), impairing the visualization of AgRP neurons. To improve our analysis, we also quantified the number of leptin-induced pSTAT3 cells in the ventromedial or lateral arcuate nucleus (ARH), where AgRP and POMC neurons are mainly located, respectively (Author response image 2 demonstrates the distinct distribution of AgRP and POMC neurons in the ARH, as well as the ARH subdivisions). We found an equivalent number of pSTAT3 cells in both ARH sub-regions of Ubi-LepRNull mice, compared to Ubi mice, indicating that the LepR restoration was homogeneous in the whole ARH (new data in Figure 1K).

**Author response image 2. respfig2:** 

AgRP neurons are also essential for the refeeding response after fasting in mice consuming standard chow (Luquet et al., 2007). In this sense, we measured the food intake 4, 12, 24 and 48 hours after a 24 hours fasting period and observed a similar refeeding response in male and female Ubi and Ubi-LepRNull mice (Author response image 3). Thus, if LepR expression had not been completely restored in AgRP neurons, the refeeding response may not be normalized as well. Additionally, hypothalamic AgRP and NPY mRNA expression were normalized in Ubi-LepRNull mice (Figure 3G).

**Author response image 3. respfig3:** 

Altogether, we feel that these results provide evidence that LepR signaling was restored in AgRP neurons, indicating that the reproductive defects exhibited by Ubi-LepRNull mice were not associated with an incomplete reactivation of LepR in AgRP neurons. We added the refeeding data (as Figure 3—figure supplement 2A, B) and the separate ARH pSTAT3 counting in the revised manuscript (Figure 1K). However, we do not feel comfortable to add the co-localization between pSTAT3 and AgRP due to the limitations mentioned earlier.

2) The ability to partially restore brain weight in adult mice is remarkable, even if it is incomplete. Since it is not known where leptin signaling is important for regulation brain weight, a more thorough analysis of Lepr signaling in specific cells is not feasible. However, it is important to know whether the increase in brain weight is due to an increase in cell number (DNA content) or edema. The former result would suggest that future investigations into what cells proliferate in the adult and where they are located would be warranted.

To investigate the possible mechanisms behind the increase in brain mass after LepR reactivation, we measured the DNA content in the hippocampus, cerebral cortex and hypothalamus, as suggested by the reviewer. Interestingly, we found an increased DNA content in the cerebral cortex of Ubi-LepRNull mice compared to Ubi mice (lean controls), whereas no significant changes were observed in other brain regions (Author response image 4).

**Author response image 4. respfig4:** 

This finding suggests that the decreased brain mass of Ubi-LepRNull mice was not caused by a lower number of cells since a higher DNA content per mg of tissue indicates higher cell density. However, this analysis does not allow us to detect changes in different cell types in the brain. It is now well-established that leptin acts not only in neurons but also in glial cells (Ahima et al., 1999; Kim et al., 2014; Djogo et al., 2016) and since these different cells have important differences in their cell body sizes, we cannot rule out that the lack or restoration of LepR could affect the number of neurons or glial cells differently. To further investigate the causes of the lower brain mass of Ubi-LepRNull mice, we also counted the number of cells in a given area of the hippocampus, cerebral cortex and hypothalamus using the nuclear marker DAPI and the neuronal perikarya and dendrite marker Neurotrace. Using DAPI as a marker we did not observe differences in the number of cells among the experimental groups (Author response images 5 and 6).

**Author response image 5. respfig5:** 

**Author response image 6. respfig6:** 

However, we observed a trend towards a decrease in the number of Neurotrace positive cells (which represents specifically neurons) in the hippocampus and ARH of Ubi-LepRNull mice, compared to Ubi mice (Author response images 7 and 8). However, these differences did not reach statistical significance.

**Author response image 7. respfig7:** 

**Author response image 8. respfig8:** 

Altogether, these quantifications also indicate that the differences in brain mass among our experimental groups were not mainly caused by changes in the number of cells, in accordance with our previous DNA content data. Although the reviewer question was very pertinent and clever, we feel that the data shown here is not conclusive without additional experiments such as stereological analysis or automatic isotropic fractionation for large-scale quantitative cell analysis (Azevedo et al., 2013). However, such measurements would be beyond the scope of the current study. Therefore, we prefer not to include the DNA and cell counting results in the revised version of the manuscript as we consider the data preliminary. Nevertheless, a paragraph discussing whether the increase in brain mass was due to an increase in cell number or edema was added in the revised manuscript:

“The remarkable increase in brain mass after LepR reactivation can be explained by an increase in cell number and/or edema. […] Therefore, additional experiments are still necessary to investigate the possible mechanisms behind the increase in brain mass after LepR reactivation.”

3) The long-standing observation that there is a critical period in early development for AgRP and POMC neuron maturation is challenged by the current results. The data in Figure 7 reveal extensive restoration of AgRP and POMC immunofluorescence in one major target (PVN) of AgRP and POMC neurons. The question is whether restoration of LepR signaling in adult mice promoted axon extension into the PVN or whether the axons were there all along, but lacked AgRP and POMC due to inadequate synthesis or trafficking. Ideally, the authors should design an experiment to visualize the AgRP and POMC axons in the PVN independently of these peptides. Alternatively, a cogent discussion of the issue is warranted.

We agree with the reviewer that staining AgRP or POMC-derived peptides not necessarily allow the visualization of ARH neuron axons. However, Bouret et al. (2004) used the anterograde tracer DiI to demonstrate that the lack of leptin signaling disrupts the normal developmental pattern of projections from the ARH to the paraventricular nucleus of the hypothalamus (PVH), already demonstrating that this neurocircuit does not fully develop without leptin signaling. To further investigate these projections, we crossed *Lep^ob/+^* mice with LepR-Cre::LSL-tdTomato mice, in order to produce leptin deficient mice expressing the tdTomato fluorescent protein only in leptin receptor cells. Importantly, tdTomato is uniformly distributed throughout the cell processes, allowing the visualization of fine structural details, such as thin axons or long-range axonal projections (Madisen et al., 2010). Thus, we compared the distribution of axons in the PVH between control (LepR-Cre::LSL-tdTomato) and *Lep^ob/ob^*::LepR-Cre::LSL-tdTomato mice. We found a significant reduction in the amount of axons from LepR-expressing neurons in the PVH of *Lep^ob/ob^* mice compared to control animals (Figure below). This finding is another evidence that the lack of leptin signaling disrupts the projections from the ARH to the PVH. Therefore, LepR restoration somehow promoted axon extension into the PVH. The *Lep^ob/ob^* data were included in the manuscript as a Figure 7—figure supplement 2 and these findings discussed in more detail in the revised manuscript.

“In our study, these projections were investigated by staining AgRP or POMC-derived peptides, which not necessarily allow the complete visualization of ARH neuron axons. […] Additionally, we found a significant reduction in the amount of axons from LepR-expressing neurons in the PVH of Lep^ob/ob^ mice compared to control animals. Thus, these findings suggest that the higher density of AgRP and POMC fibers in the PVH after LepR restoration was likely caused by axonal growth.”

Reviewer #1:This paper represents a comprehensive study of the consequences of restoring leptin receptor function in mice with a null mutation in that receptor gene on many aspects of physiology, anatomy and behavior. The restoration was either done at 4 weeks (before obesity) or at 10 weeks, when the mice were already quite obese. Virtually all parameters that were measured improved after restoration of the LepR, but in some cases there were still small differences compared to normal littermates. Those differences presumably represent permanent changes that cannot be fully reversed, e.g. brain weight can't be restored in the absence of neurogenesis. Fertility is one parameter that was not fully restored, whereas previous studies have suggested that it can be restored with expression of LepR only in AgRP neurons and ablation of AgRP neurons in Lep ob/ob mice also restores fertility. Thus, a more thorough analysis of LepR activity in AgRP neurons is warranted because Ubc-Cre may not be fully functional in those critical neurons. The current results contradict previous studies indicating that leptin signaling is required during a critical period to promote axon outgrowth (or neuropeptide accumulation) in target tissues. The results presented here reveal that neuropeptide immunostaining is fully restored even after 10 weeks without leptin signaling. Bolstering this result with additional approaches would be a valuable addition. It seems unlikely that growth of new axons (fibers) into target tissues actually occurs after restoring leptin signaling. A more likely possibility is that the fibers are there but not visible due to low peptide abundance. Distinguishing between these possibilities would be a welcome addition.

We would like to thank the reviewer for his/her important considerations and suggestions. In our previous response, we provided additional data indicating that the lack of leptin signaling during development disrupts the projections from the ARH to the PVH. Therefore, it is not a matter of not being able to visualize these fibers after staining AgRP or POMC-derived peptides. Consequently, LepR restoration in adult animals somehow promoted axon extension into the PVH.

Reviewer #2:[…] The described restoration of hypothalamic projections appears difficult to comprehend, i.e. that neuron send projections after 10 weeks of neurocircuitry development. This clearly represents a very interesting finding. Nevertheless, currently this is based on staining for endogenous peptides whose expression is altered in the models. Thus, complementary experiments with genetically marking anatomical projections independent of the endogenously expressed peptides should be performed to substantiate this finding.

We would like to thank the reviewer for his/her important considerations. We agree with the reviewer that staining AgRP or POMC-derived peptides not necessarily allow the visualization of ARH neuron axons. However, Bouret et al., 2004, used the anterograde tracer DiI to demonstrate that the lack of leptin signaling disrupts the normal developmental pattern of projections from the arcuate nucleus (ARH) to the paraventricular nucleus of the hypothalamus (PVH), already demonstrating that this neurocircuit does not fully develop without leptin signaling. To further investigate these projections, we crossed *Lep^ob/+^* mice with LepR-Cre::LSL-tdTomato mice, in order to produce leptin deficient mice expressing the tdTomato fluorescent protein only in leptin receptor cells. Importantly, tdTomato is uniformly distributed throughout the cell processes, allowing the visualization of fine structural details, such as thin axons or long-range axonal projections (Madisen et al., 2010). Thus, we compared the distribution of axons in the PVH between control (LepR-Cre::LSL-tdTomato) and *Lep^ob/ob^*::LepR-Cre::LSL-tdTomato mice. We found a significant reduction in the amount of axons from LepR-expressing neurons in the PVH of *Lep^ob/ob^* mice compared to control animals. Therefore, it is not a matter of not being able to visualize these fibers after staining AgRP or POMC-derived peptides. Consequently, LepR restoration in adult animals somehow promoted axon extension into the PVH. The *Lep^ob/ob^* data were included in the manuscript as Figure 7—figure supplement 2 and these findings discussed in more detail in the revised manuscript (Discussion, tenth paragraph).

Reviewer #3:[…] What the authors can conclude is that the restoration of Lepr signaling achieved at 10 weeks in their system largely corrects body weight, feeding, energy expenditure phenotypes but has much more modest effects on reproductive and neuroanatomical phenotypes. This qualitative difference in phenotype restoration may be due to a permanent developmental defect or to a greater degree of sensitivity to leptin signaling of certain phenotypes.The authors do acknowledge this shortcoming. It might be helpful to quantify the restoration of LepR expression in populations of neurons to determine efficiency. However, there is an inherent challenge of any system in which restoration of genetic mutation is not 100% efficient and one does not achieve a complete restoration of wild-type phenotypes.

We would like to thank the reviewer for his/her important considerations and suggestions. We used several approaches to confirm the efficacy of LepR restoration in our model, including: 1) assessment of *Lepr* mRNA expression; 2) counting the number of pSTAT3 immunoreactive cells after an acute leptin injection in critical brain areas that mediate leptin’s effects; 3) evaluation of the anorexigenic effects induced by leptin treatment. In all these measurements, Ubi-LepRNull mice exhibited equivalent values as those found in lean controls (Ubi mice). Additionally, although we observed several defects in the Ubi-LepRNull mice, we also observed a complete normalization in important parameters, such as in food intake, respiratory quotient, AgRP/NPY hypothalamic expression, thyroid and adrenal axes, memory and arcuate neural projections to the paraventricular nucleus. The complete recovery in these outcomes is also indirect evidence that the LepR restoration achieved was sufficient to test our hypothesis. Finally, as pointed out by the reviewer, we also acknowledged in the Discussion section that it is possible that part of phenotype observed was caused by an incomplete reactivation that could not be detected by the methods employed in the present study.